# Homologous heteropolyaromatic covalent organic frameworks for enhancing photocatalytic hydrogen peroxide production and aerobic oxidation

Jiani Yang[1], Zhenyang Zhao[1], Xu Liu[1], Heng Yang[1], Jin Yang[2], Mi Zhou[1], Shuang Li [1], Xikui Liu [1] ✉, Xiaohui Xu [3] ✉ & Chong Cheng [1,4,5] ✉

The development of robust catalysts that can work under harsh conditions bring promise but a challenge for photocatalytic hydrogen peroxide production. Here, we report the design of thiazole-based homologous heteropolyaromatic COFs (TTT-COF) via post-cyclization reaction for photocatalytic $H_2O_2$ production and aerobic oxidation of $C(sp^3)$-H bonds. Our studies demonstrate that the elemental S heteroatom enables modified COF materials with high chemical stability, continuous π-conjugation, efficient electron and energy transfer, and an enhanced donor-acceptor (D-A) structure and charge separation, thus boosting their intrinsic photocatalytic activities and stability. Consequently, TTT-COF achieves a photosynthetic $H_2O_2$ production rate of 29.9 mmol $g^{-1}$ $h^{-1}$ with more than 200 hours of long-term stability when employing 10 % benzyl alcohol (V/V) as a sacrificial agent. Notably, the TTT-COF photocatalyst exhibits high reactivity in the oxidation of ethylbenzene derivatives. We believe this strategy offers a promising pathway to synthesize homologous heteropolyaromatic COFs and holds the potential for large-scale production of COF materials with tailored properties for broad applications in photocatalysis and beyond.

Hydrogen peroxide ($H_2O_2$) stands as a cornerstone chemical in the industrial landscape, with its applications spanning disinfection, bleaching, chemical synthesis, and aerospace[1,2]. Its high oxidative potential under benign conditions, coupled with the environmentally benign by-products of water and oxygen, renders it an attractive oxidant. However, the predominant industrial synthesis of $H_2O_2$— accounting for over 95% of global production—relies on the anthraquinone oxidation process[3], which is energy-intensive and employs noble-metal catalysts and substantial volumes of organic solvents under high pressure[4–6]. Consequently, there is a pressing need for alternative, green, and sustainable pathways for $H_2O_2$ generation. Photocatalytic production of $H_2O_2$ under mild conditions presents itself as a viable and environmentally friendly alternative[7–13]. In this process, light irradiation triggers the generation of oxygen-containing intermediates such as superoxide (•$O_2^-$), singlet oxygen ($^1O_2$), or hydroxyl radicals (•OH)[14,15]. The field has been invigorated by pioneering studies that have explored a range of semiconductors for photocatalytic $H_2O_2$ synthesis and aerobic oxidation. Among the

[1]College of Polymer Science and Engineering, State Key Laboratory of Advanced Polymer Materials, Sichuan University, Chengdu, China. [2]Department of Chemistry, Chemistry Research Laboratory, University of Oxford, Oxford, UK. [3]Department of Ultrasound, Frontiers Science Center for Disease-Related Molecular Network, West China Hospital, Sichuan University, Chengdu, China. [4]Institute of Chemistry and Biochemistry, Free University of Berlin, Takustr. 3, Berlin, Germany. [5]Department of Endodontics, State Key Laboratory of Oral Diseases, National Center for Stomatology, West China Hospital of Stomatology, Sichuan University, Chengdu, China. ✉e-mail: xkliu@scu.edu.cn; xiaohuixu@scu.edu.cn; chong.cheng@scu.edu.cn

leading contenders are inorganic nanomaterials, such as titanium dioxide (TiO$_2$)[16], zinc oxide (ZnO)[10], or cadmium sulfide (CdS)[17], as well as heavily reported organic polymers and frameworks[18–20] in recent years that include linear conjugated polymers[21–24], graphitic carbon nitride (g-C$_3$N$_4$)[25,26], covalent triazine frameworks (CTFs)[27], and covalent organic frameworks (COFs)[28].

Polyaromatic covalent organic frameworks (COFs) represent a burgeoning class of crystalline porous polymers, defined by their lightweight organic backbones interconnected through covalent bonds[2,29–38], which have attracted intensive attention due to their tunable bandgap, abundant catalytic sites[39], and remarkable photochemical stability, making it a popular choice for photocatalytic reactions[40–44]. However, the current landscape of synthetic methods for COF is diverse, with a plethora of structural units and a wide array of bonding modes[45–54]. While offering a rich tapestry of possibilities, the diversity of COF-based photocatalysts presents a formidable challenge in analyzing the corresponding relationships between the chemical structures and photocatalytic performances. In contrast, homologous polyaromatic COFs refer to a class of COFs constructed from polycyclic aromatic hydrocarbon building units/monomers and bonding modes that share structural homology[55]. Thus, it is suggested that the high degree of structural similarity and uniformity of homologous polyaromatic COFs will facilitate a more coherent analysis of the structure and function of their underlying photocatalytic mechanisms[56].

While the current synthesis strategies of homologous polyaromatic COFs are mostly based on imine-linked frameworks[57–59], which have resulted in several challenges for enhancing their photocatalytic properties in broad applications: 1) the reversibility of imine bonds poses a challenge to their chemical stability[60–62]; 2) the strong polarization of imine linkages impedes π-electron delocalization, resulting in discontinuous π-conjugation within COFs and impedes energy transfer (EnT) process[63–69]; 3) homology of molecular frameworks often leads to a reduction in the donor-acceptor (D-A) structure and charge separation[70]. To address these challenges, incorporating functional molecule units into homologous COFs via chemical locking of the imine linkage has been recognized to be an efficient strategy for transforming reversible linkages into stable and functional covalent bonds in COFs[71–75]. As an example, Thomas's group has reported the post-functionalization of COFs using the multicomponent Povarov reaction or the three-component Doebner reaction to yield stable quinoline-linked COFs for enhanced H$_2$O$_2$ photosynthesis[70,76]. More recently, the same group also proposed the multicomponent domino and Povarov reactions to design the heteropolyaromatic COF[55]. Though recent research has proposed synthesizing the thiazole-based homologous polyaromatic COFs via sulfur vapor strategy[77,78], the effects and mechanisms of the post-cyclization reaction via sublimated sulfur atoms on the optical/optoelectronic properties and corresponding photocatalytic applications of these homologous polyaromatic COFs have not been disclosed.

Here, we propose the design of thiazole-based homologous heteropolyaromatic COFs (TTT-COF) via a solvent-free and eco-friendly post-cyclization reaction for enhancing photocatalytic H$_2$O$_2$ production and the aerobic oxidation of C(sp$^3$)-H bonds. As illustrated in Fig. 1, our research is driven by three primary purposes: 1) introducing the S heteroatom to form heteropolyaromatic COF enables remarkable chemical stability (Fig. 1a); 2) the unique structure with continuous π-conjugation of TTT-COF guarantees efficient single electron transfer (SET) and energy transfer (EnT) for photocatalysis; 3) the thiazole-based frameworks can significantly enhance the intramolecular polarity, D-A structure, and charge separation (Fig. 1b, c), thus boosting the intrinsic photocatalytic activities[70]. As a result, compared to its imine-linked COF (TTI-COF), TTT-COF exhibits high chemical stability in harsh environments, high conjugation in the molecular plane while capturing oxygen-containing intermediates, and better photocatalytic

and optoelectronic properties. Consequently, TTT-COF achieves a photosynthetic H$_2$O$_2$ production rate of 29.9 mmol g$^{-1}$ h$^{-1}$ when employing 10 % benzyl alcohol (V/V) as a sacrificial agent, which is a 20-fold improvement over the original imine-linked COF (1.4 mmol g$^{-1}$ h$^{-1}$) and exceeds many of the state-of-the-art organic and inorganic photocatalysts. Its solar-to-chemical conversion efficiency for H$_2$O$_2$ production reaches 0.32%, exceeding the global average of natural photosynthetic organisms (-0.10%)[79]. Notably, TTT-COF maintains long-term photocatalytic stability that can operate continuously for more than 200 hours without a significant decline in activity. More importantly, benefiting from the continuous π-conjugation structure, the constructed photocatalyst TTT-COF can undergo rapid intermolecular migration of photogenerated charges and substrate (C(sp$^3$)-H bonds) activation under light irradiation to achieve high reactivity for the oxidation of ethylbenzene derivatives. This strategy not only promises the universal, solvent-free, and eco-friendly synthesis of homologous heteropolyaromatic COFs but also holds the potential for large-scale production of COF materials with tailored properties for a range of applications in photocatalysis and beyond.

## Results
### Synthesis and characterization of COFs
We initiated our synthesis with the imine-based TTI-COF, prepared via solvothermal condensation of tris(4-formylphenyl) triazine and tris(4-aminophenyl) triazine (see the Methods for details). After a reaction for 72 hours at 120 °C, a yellowish powder was obtained, representing the TTI-COF (Fig. 1a). The subsequent step involved the transformation of TTI-COF into a thiazole-based homologous heteropolyaromatic COF (TTT-COF) through a post-cyclization reaction with elemental sulfur. This process commences with the oxidation of aromatic imines by sublimated sulfur vapor at 350 °C, yielding thioamides, which then undergo an oxidative cyclization reaction to form thiazole rings (Supplementary Fig. 1)[78]. Here, molecular sulfur (S$_8$) dually functions as an oxidizing agent and a nucleophile, initially binding to the imine carbon and subsequently to the phenyl ring adjacent to the nitrogen of the imine, which offers a way to create homologous heteropolyaromatic COF with tailored electronic properties and enhanced stability.

The crystalline nature and structural frameworks of TTI-COF and TTT-COF were meticulously examined through powder X-ray diffraction (PXRD) analyses. Prominent and sharp peaks in the PXRD patterns of both materials are indicative of their high degree of crystallinity. To delineate their lattice parameters, Pawley refinements were meticulously applied to the experimental PXRD data (Fig. 2a, b). Both COFs were found to crystallize in the same hexagonal P-6 space group, with an AA stacking model, yielding satisfactory fits (Supplementary Figs. 2-7). A subtle contraction in the lattice parameter '*a*' from 25.81 Å in TTI-COF to 25.15 Å in TTT-COF was observed, attributable to the oxidative cyclization process and the consequent reduction in pore size (Supplementary Tables 1,2).

The porous characteristics of TTI-COF and TTT-COF were further evaluated using nitrogen adsorption isotherms, as depicted in Fig. 2c. The Brunauer-Emmett-Teller (BET) surface areas of TTI-COF and TTT-COF were measured and found to be 1368 m$^2$ g$^{-1}$ and 1472 m$^2$ g$^{-1}$, respectively. Pore size distribution analyses, employing the nonlocal density functional theory model, revealed average pore diameters of approximately 2.3 nm for TTI-COF and 1.9 nm for TTT-COF. Furthermore, the periodic hexagonal architectures of TTI-COF and TTT-COF were visualized using high-resolution transmission electron microscopy (HRTEM). By aligning the 2D COF materials perpendicular to the incident electron beam, the hexagonal pores were directly imaged along the channel direction (*c* axis) (Fig. 2d, e). These magnified views corroborated the structural and pore size dimensions inferred from the crystallographic data. Additionally, elemental dispersive spectroscopy (EDS) mapping confirmed the uniform distribution of carbon, nitrogen, and sulfur across the COF matrix (Supplementary Figs. 8, 9),

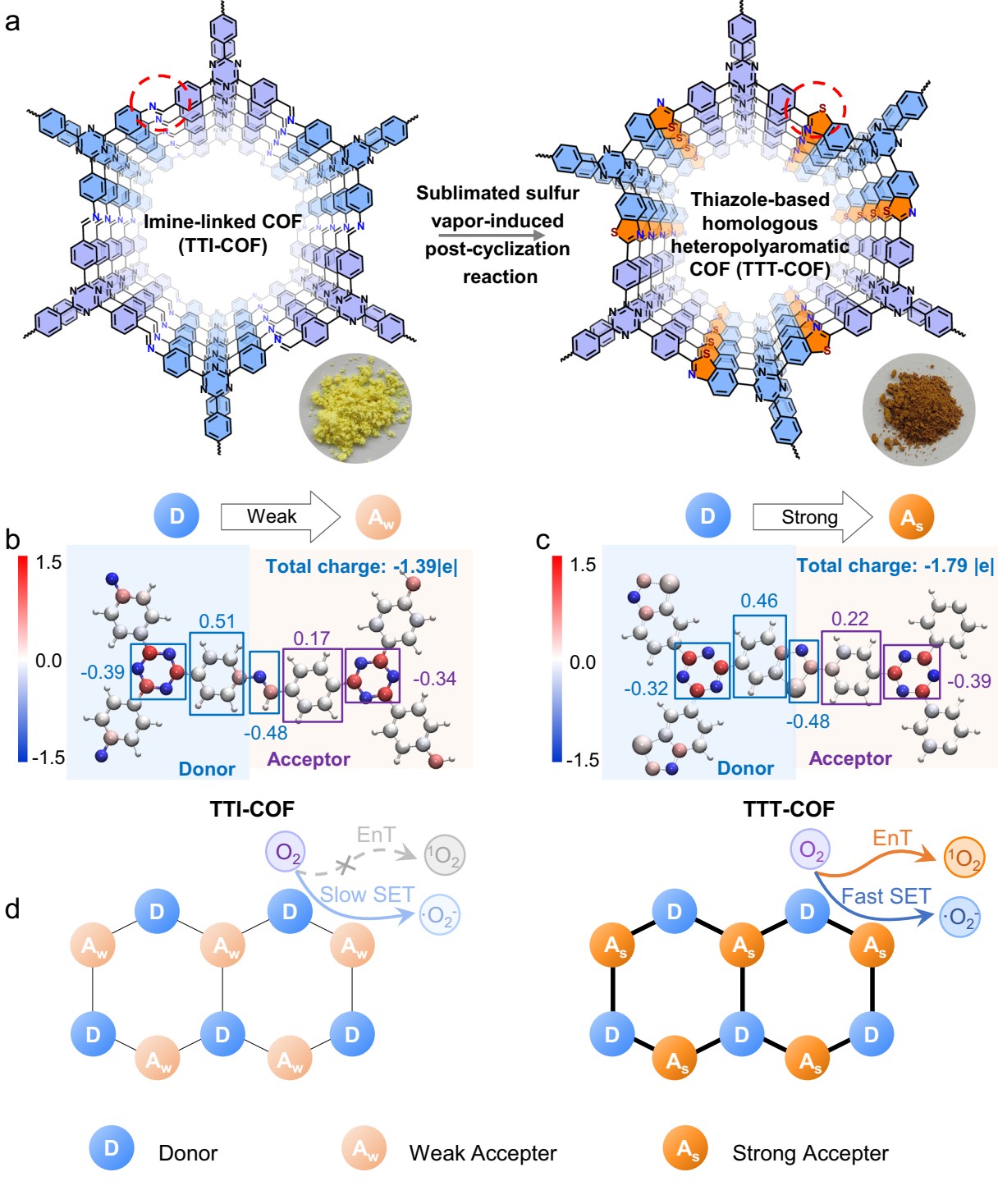

**Fig. 1 | Design of homologous heteropolyaromatic COFs via post-cyclization reaction. a** Synthesis of triphenyl triazine thiazole COF (TTT-COF) from triphenyl triazine imine COF (TTI-COF) by the post-cyclization reaction from sublimated sulfur vapor, the inserted images are the synthesized COFs. **b** The Bader charge coloring distribution of TTI-COF. **c** Bader charge coloring distribution of TTT-COF.

Atom colors: C, white; N, blue; O, red; S, gray; H, small white balls. Specific colors: blue (−1.5), white (0.0), red (1.5), and the middle color is defined according to the corresponding value (Bader value). **d** The schematic illustration of the structural advantages of homologous heteropolyaromatic COFs (TTT-COF).

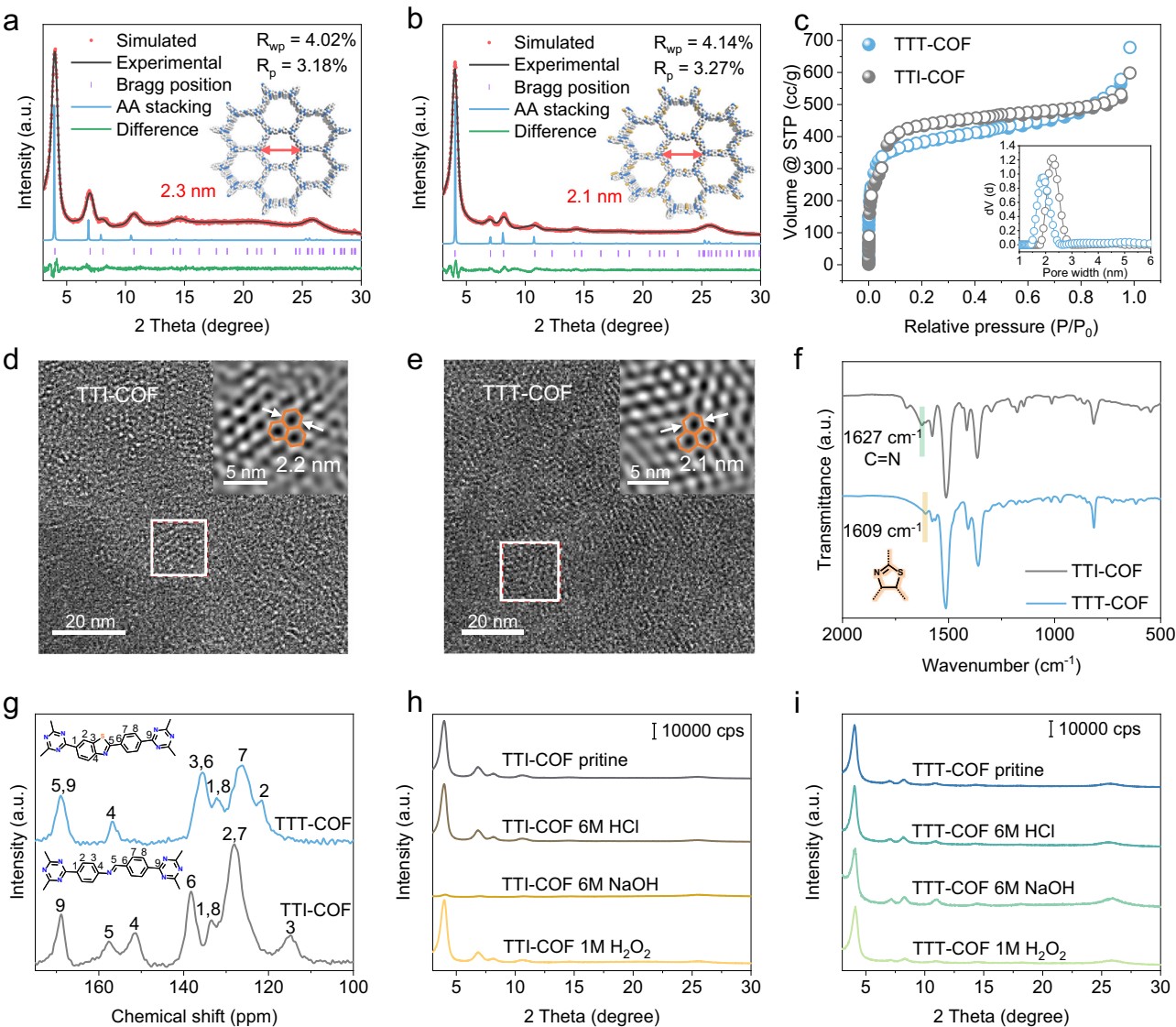

**Fig. 2 | Chemical structure properties of photocatalytic COFs.** Pawley refinements against the experimental PXRD patterns of (**a**) TTI-COF and (**b**) TTT-COF. Experimental (black line), Pawley fitted (blue line), simulated (red dot), and difference (green line) are displayed. Bragg positions are marked by purple bars. Atom colors: C, white; N, blue; S, orange. **c** N$_2$ adsorption−desorption isotherms of TTI-COF and TTT-COF, and the inset is pore size distributions. The HRTEM images of TTI-COF (**d**) and TTT-COF (**e**) (inset is taken from the section highlighted in **d** and **e**). **f** FTIR spectra and **g** $^{13}$C ssNMR data of TTI-COF and TTT-COF. Chemical stability tests of TTI-COF (**h**) and TTT-COF (**i**) after 1 day of treatment with 6 M HCl, 6 M NaOH, and 1.0 M H$_2$O$_2$. Experiments were repeated independently (**a**–**i**) three times with similar results. In **a**, **b**, **f**, **g**, **h**, **i**, a.u. indicates the arbitrary units. Source data are provided as a Source Data file.

attesting to the homogeneous integration of sulfur within the COF scaffold. This comprehensive structural characterization underscores the successful synthesis of TTT-COF, setting the stage for its performance in photocatalytic applications.

The successful synthesis of TTI-COF, characterized by the imine linkage, was initially confirmed by Fourier-transform infrared spectroscopy (FTIR). The distinctive imine (N = CH) vibration at 1627 cm$^{-1}$ provided clear evidence of the imine formation (Fig. 2f and Supplementary Tables 3, 4). The transformation of TTI-COF to TTT-COF was further substantiated by the vanishing of the imine vibration and the emergence of a new N = C vibration characteristic of the thiazole ring at 1609 cm$^{-1}$. Elemental analysis shows the presence of sulfur with an elemental composition close to the composition that would be expected from the thiazole model (Supplementary Table 5). The solid-state nuclear magnetic resonance (ssNMR) spectra of TTT-COF exhibited marked and well-resolved alterations compared to its TTI-COF precursor (Fig. 2g). In the $^{13}$C NMR spectra of TTI-COF, distinct

peaks at 151 ppm and 115 ppm were assignable to the carbon atom bonded to the imine-N and the adjacent ortho aromatic carbon, respectively. After the post-cyclization reaction, the signal of the carbon atom linked to the imine-N was observed to shift to a lower frequency at 156 ppm in TTT-COF, signifying the successful conversion of the imine to the thiazole ring. Furthermore, to optimize the cyclization efficiency of TTT-COF, TTI-COF samples were subjected to treatments with varying sulfur dosages. The experimental results show that the optimal sulfur dosage for cyclization is determined to be 15 times the mass of TTI-COF (Supplementary Figs. 10, 11).

X-ray photoelectron spectroscopy (XPS) spectra further corroborated the establishment of the thiazole linkage in TTT-COF (Supplementary Figs. 12, 13). One of the advantages of thiazole linkages is their enhanced chemical stability over imine linkages. To exemplify the chemical robustness of TTT-COF, we subjected the activated samples to stringent tests in aqueous solutions: 6 M HCl (pH < 0), 6 M NaOH (pH > 14), and 1 M H$_2$O$_2$ (a potent oxidizing agent). After a 12-

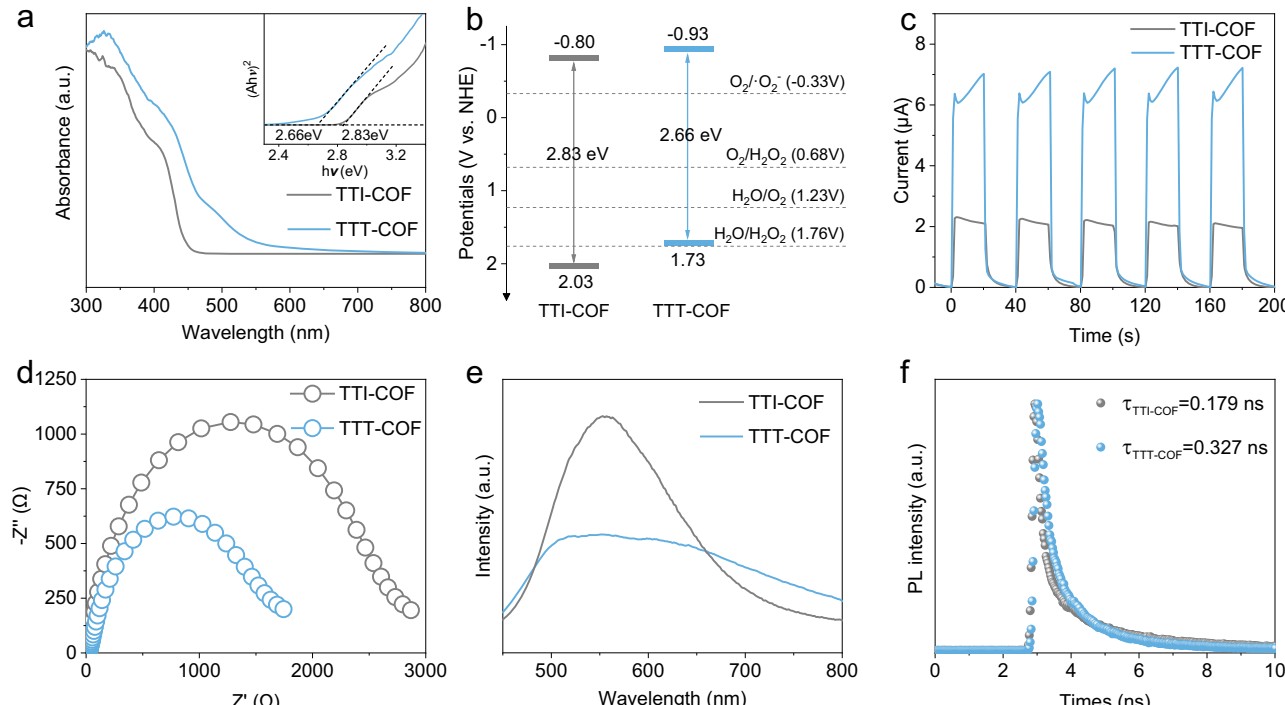

**Fig. 3 | Optical and optoelectronic properties for photocatalytic COFs. a** Solid-state UV-Vis diffuse reflectance spectrum and Tauc plot for band gap calculation. **b** Band-structure diagrams of TTI-COF and TTT-COF. **c** Photocurrent-responses of TTI-COF and TTT-COF. **d** Nyquist plots of TTI-COF and TTT-COF. **e** PL spectra of TTI-COF and TTT-COF. **f** Time-resolved fluorescence spectroscopy of TTI-COF and TTT-COF excited at 370 nm. Experiments were repeated independently (**a**, **c**–**f**) three times with similar results. In **a**, **e**, **f**, a.u. indicates the arbitrary units. Source data are provided as a Source Data file.

hour exposure, PXRD patterns with peaks as sharp as those from the pristine samples were retained, attesting to the unchanged crystallinity of TTT-COF post-immersion. In stark contrast, TTI-COF maintained only partial crystallinity upon testing in 6 M NaOH (Fig. 2h, i), underscoring the stability imparted by the thiazole rings in TTT-COF. Moreover, the thermogravimetric analysis experiment exhibits that TTT-COF has better thermal stability up to ~600 °C compared to TTI-COF (Supplementary Fig. 14). This detailed characterization not only validates the successful synthesis and structural integrity of homologous heteropolyaromatic TTT-COF but also highlights its remarkable chemical and thermal stability, a critical attribute for applications in harsh chemical environments and photocatalytic processes.

**Optical and optoelectronic characterization**

Following the confirmation of the synthesis of TTI-COF and TTT-COF through comprehensive characterization, we proceeded to investigate their light absorption properties and energy band structures. UV−vis diffuse reflectance spectra (DRS) revealed effective light absorption in the visible spectrum for both materials (Fig. 3a). Notably, TTT-COF exhibited a more extensive absorption range from 450 nm to 600 nm, attributable to its enhanced conjugation afforded by the thiazole linkages. This absorption profile corresponded well with the yellowish hue of TTI-COF and the brownish tint of TTT-COF as disclosed in Fig. 1a. The optical band gaps ($E_g$) were determined to be 2.83 eV for TTI-COF and were slightly reduced to 2.66 eV for TTT-COF, as derived from Tauc plots using the Kubelka−Munk equation from DRS analysis (Fig. 3a). The flat band potentials were ascertained to be -1.00 V for TTI-COF and -1.13 V for TTT-COF using Mott−Schottky plots (Supplementary Fig. 15). By integrating the optical band gaps with the Mott−Schottky data, the band structure configurations were deduced (Fig. 3b). The conduction bands (CB) of TTI-COF and TTT-COF were calculated to be -0.80 V and -0.93 V (vs. normal hydrogen electrode,

NHE), respectively, based on the flat band potentials. Subsequently, the valence bands (VB) were estimated to be 2.03 V and 1.73 V (vs. NHE) using the relationship $E_{CB} = E_{VB} - E_g$. Given that the CB positions of both TTI-COF and TTT-COF are more negative than the redox potentials for the 2e− oxygen reduction reaction (ORR, -0.33 V for indirect ORR, and 0.68 V for direct ORR at pH = 0, vs. NHE), they are thermodynamically capable of producing $H_2O_2$ via either a direct or indirect 2e− ORR pathway.

The dynamics of photogenerated charge separation and migration in TTI-COF and TTT-COF were examined through transient photocurrent response measurements. Both materials displayed continuous and robust photocurrent responses during alternating light exposure (Fig. 3c). TTT-COF exhibited a significantly higher photocurrent than TTI-COF, indicative of its high efficiency in photoinduced charge separation and transport. Furthermore, the charge migration resistance was assessed using electrochemical impedance spectroscopy (EIS), with TTT-COF showing a smaller semicircular radius in the Nyquist plot, signifying faster charge transport (Fig. 3d). The steady-state photoluminescence (PL) spectra revealed pronounced quenching in TTT-COF compared with TTI-COF (Fig. 3e), suggesting reduced charge recombination in the former. Time-resolved fluorescence spectroscopy, highly sensitive to photogenerated charge carriers, was employed to collect fluorescence decay curves with a 370 nm pump (Fig. 3f). The average fluorescence lifetimes were determined to be 0.179 ns for TTI-COF and 0.327 ns for TTT-COF, with the longer lifetime of TTT-COF indicating slower recombination of photoinduced charge carriers, consistent with the steady-state PL results. The enhanced charge separation and transfer efficiency in TTT-COF are attributed to the formation of a homologous heteropolyaromatic structure with thiazole rings, which augments the overall conjugation of the framework and creates new electron donor-acceptor sites, thereby facilitating efficient photocatalytic performance.

## Photocatalytic $H_2O_2$ production

We conducted photocatalytic experiments under ambient conditions at 25 °C, utilizing a sacrificial electron donor mixture (9:1 v/v water/benzyl alcohol (BA)) and continuously purging with $O_2$. A xenon lamp emitting light with wavelengths greater than 420 nm was employed to irradiate the reaction mixture, and $H_2O_2$ production was quantified using iodometric titration (see Methods for details and Supplementary Figs. 16, 17). Both TTI-COF and TTT-COF exhibited a distinct linear correlation between $H_2O_2$ generation and exposure to light, underscoring their photocatalytic competence. The reaction temperature of the post-cyclization reaction, pivotal in the transformation of imine to thiazole linkages, was optimized by preparing a series of samples subjected to various temperatures ranging from 150 °C to 450 °C (Supplementary Fig. 18). The photocatalytic performance varied significantly with the temperature, with the optimal conversion achieved at 350 °C, yielding a photocatalytic $H_2O_2$ production rate of 29.9 mmol g$^{-1}$ h$^{-1}$. This outcome signifies the complete conversion of imine to thiazole while preserving the integrity of the COF's reticular structure. In stark contrast, TTI-COF demonstrated a markedly lower production rate of 1.4 mmol g$^{-1}$ h$^{-1}$, underscoring the pivotal role of thiazole linkages in enhancing photocatalytic efficiency (Fig. 4a). In addition, it is found that the photocatalytic performances of different TTT-COF samples are related to the treatment conditions with varying sulfur dosages. The fully cyclized COFs (0.5eq-15eq) exhibited similar $H_2O_2$ production, confirming the critical role of complete cyclization in photocatalytic performance (Fig. 4b).

It is noteworthy that the homologous heteropolyaromatic COF is capable of producing $H_2O_2$ directly from air and water, obviating the need for sacrificial reagents. This capability is important, as it facilitates scalable and practical photosynthesis with reduced costs and simplified procedures. This is achieved by scavenging photogenerated holes with water molecules through a water oxidation reaction rather than relying on organic sacrificial reagents. In pure water under air, the $H_2O_2$ production rates were measured to be 0.48 mmol g$^{-1}$ h$^{-1}$ for TTI-COF and 3.71 mmol g$^{-1}$ h$^{-1}$ for TTT-COF. When the system was aerated with $O_2$, the rates increased to 0.52 mmol g$^{-1}$ h$^{-1}$ and 4.75 mmol g$^{-1}$ h$^{-1}$, respectively. Notably, when $O_2$ was replaced with $N_2$, the photoreaction ceased, and no $H_2O_2$ was produced (Fig. 4c). Moreover, our photocatalysts demonstrated versatility across various conditions. The photocatalysis experiments of TTT-COF under different sacrificial agents showed that BA performed best as a sacrificial agent, indicating that a two-phase reaction system composed of water and BA could strongly suppress the $H_2O_2$ decomposition because the active sites of COFs remain in the BA phase whereas the formed $H_2O_2$ quickly diffuses into the water (Supplementary Fig. 19). The apparent quantum yields (AQY) values were determined to be 12% for TTT-COF and 3.9% for TTI-COF at 400 nm in a water/BA mixture under an $O_2$ atmosphere (Fig. 4d and Supplementary Fig. 20). Furthermore, the solar chemical conversion (SCC) efficiency of TTT-COF reached 0.32%, surpassing the typical photosynthetic efficiency of plants, which stands at 0.10%. These findings demonstrate the remarkable photocatalytic prowess of TTT-COF, highlighting its potential for applications in artificial photosynthesis and environmental remediation, where efficient $H_2O_2$ generation is of paramount importance.

The homologous heteropolyaromatic TTT-COF showcased long-term photocatalytic stability, a hallmark of its robustness. In a continuous operational test, TTT-COF maintained a steady $H_2O_2$ production rate of ~4 mM h$^{-1}$ over an extended period of 200 hours, with no significant degradation in catalytic activity (Fig. 4e). By the end of this test, the $H_2O_2$ concentration in the solution reaches 0.735 M, a level that is considered practical for broad applications. To simulate more realistic environmental conditions, a long-term $H_2O_2$ production test was conducted using natural sunlight as the energy source and seawater and air as reactants. A steady production of $H_2O_2$ was observed, reaching 3.60 mmol g$^{-1}$ after 9 hours (Supplementary Fig. 21). The

contribution of the catalyst amount to the $H_2O_2$ concentration revealed volcano-type curves with a maximum $H_2O_2$ production rate of 6.3 mM h$^{-1}$ at the catalyst amount of 20 mg and pH = 7, suggesting the balanced generation and decomposition of $H_2O_2$ (Supplementary Fig. 22). Meanwhile, TTT-COF maintained a consistent photosynthesis rate in aqueous solutions across different pH values. In contrast, the $H_2O_2$ production rate of TTI-COF varied from 1.4 mmol g$^{-1}$ h$^{-1}$ at pH = 7 to 2.6 mmol g$^{-1}$ h$^{-1}$ at pH = 3, due to the protonation of the imine linkages (Supplementary Fig. 23).

This performance underscores the advantage of heterogeneous catalysts over their homogeneous counterparts, particularly in terms of recyclability. To assess this, we subjected the COFs to four consecutive cycles, each comprising a 1-hour $H_2O_2$ production run (Fig. 4f, h). A precipitous decline in the $H_2O_2$ production rate was observed for TTI-COF after only two cycles, concomitant with a partial loss of its crystalline structure, as evidenced by the weakened diffraction peaks in the PXRD pattern. In stark contrast, TTT-COF exhibited a negligible loss in production rate across the four cycles, with its crystallinity preserved intact. Pawley refinements on the PXRD data post-four catalytic cycles, performed using the same space group as the pristine COF, confirmed that the unit cell parameters were virtually unchanged, closely resembling those of the as-synthesized sample (Supplementary Fig. 24 and Supplementary Table 6). This finding attests to the remarkable stability inherent in the TTT-COF. Furthermore, the chemical composition of TTT-COF remained unaltered after four cycles, as evidenced by FTIR and XPS analyses (Supplementary Figs. 25, 26).

These results collectively demonstrate the enduring stability and recyclability of TTT-COF, reinforcing its potential as a promising photocatalyst for sustainable $H_2O_2$ production. A comparative analysis of the $H_2O_2$ production performance of TTT-COF with various state-of-the-art photocatalysts reported in recent years is presented in Fig. 4g. TTT-COF presents superior photocatalytic activities in terms of both production rates and AQY values, surpassing the majority of $H_2O_2$-generating photocatalysts documented to date (Supplementary Table 7). Additionally, we also evaluated the photocatalytic antibacterial efficacy of TTT-COF since $H_2O_2$ has been widely used for bacterial disinfection. The data suggested that after 30 minutes of light exposure, the antibacterial rate impressively exceeded 99%, which may offer a safe and potent strategy against antibiotic-resistant pathogens (Supplementary Figs. 27-29).

Furthermore, the proposed oxidative cyclization synthetic strategy is a universal modification strategy. Here, two additional types of COF were prepared to demonstrate that the sulfur addition reaction can also convert the imine linkages to thiazole linkages in other COFs, thereby enhancing their photocatalytic performances. For instance, triazine phenylenediamine imine COF (TPI-COF) and tris(4-formlphenyl) benzene and tris(4-aminophenyl)-benzene (BBI-COF) were synthesized via solvothermal reactions, followed by the preparation of TPT-COF (triazine phenylenediamine thiazole COF) and BBT-COF (benzene benzene thiazole COF) under similar conditions as those of TTT-COF (See Methods for details). Both FTIR and $^{13}$C ssNMR analyses confirmed the successful conversion of imine to thiazole linkages (Supplementary Figs. 30-41). In an aqueous solution of water/BA and under an $O_2$ atmosphere, TPT-COF and BBT-COF achieved an impressive $H_2O_2$ yield of 22.7 mmol g$^{-1}$ h$^{-1}$ and 17.6 mmol g$^{-1}$ h$^{-1}$, respectively, which are approximately 20 times and 21 times higher than that of TPI-COF (1.1 mmol g$^{-1}$ h$^{-1}$) and BBI-COF (0.9 mmol g$^{-1}$ h$^{-1}$) (Supplementary Fig. 42). This result underscores the substantial improvement in photocatalytic $H_2O_2$ production performance conferred by the formation of heteropolyaromatic thiazole bonds.

## Photocatalytic mechanisms and pathways

To elucidate the photocatalytic mechanism underlying $H_2O_2$ production in this study, we conducted a series of experiments, including the

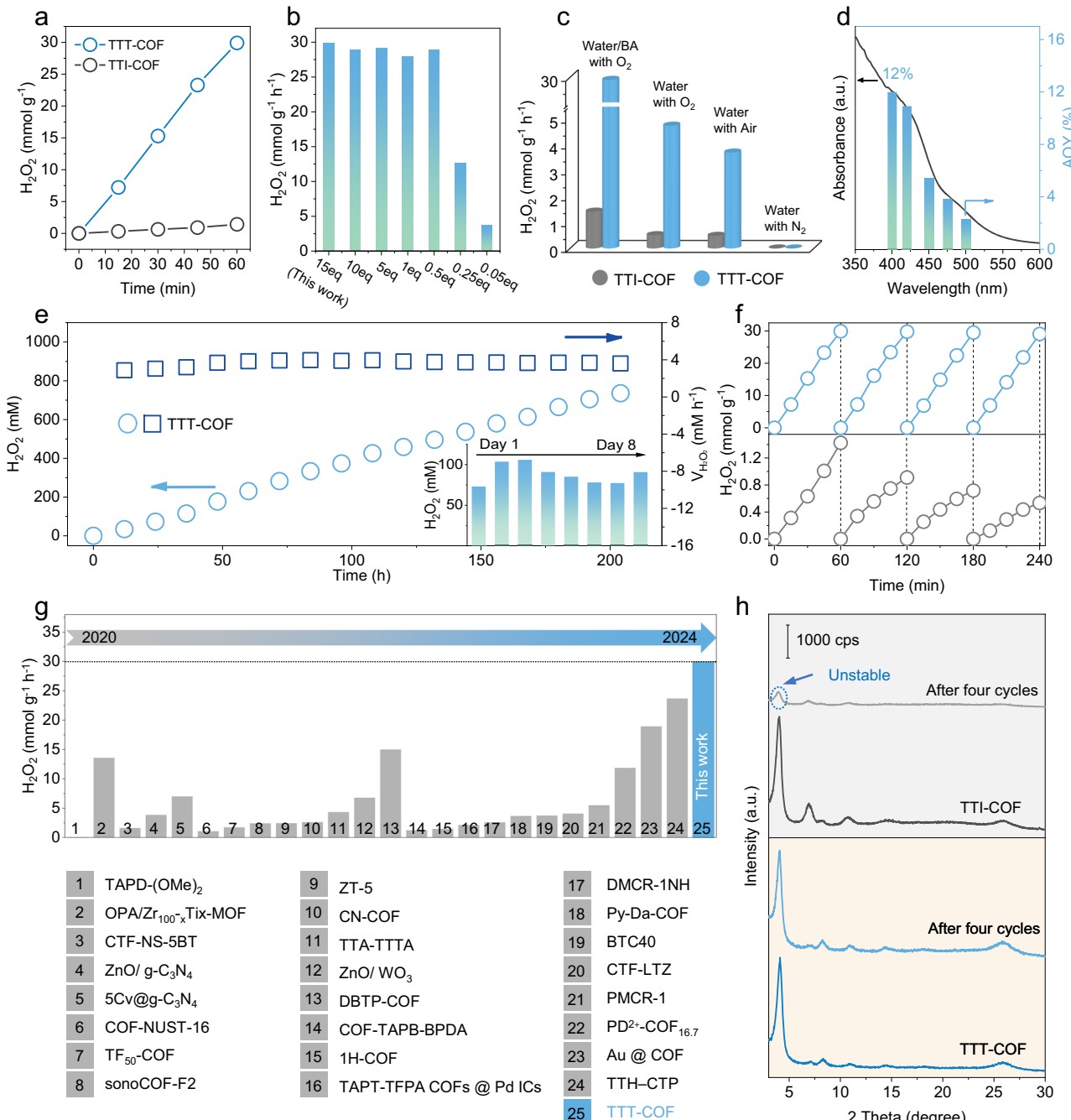

**Fig. 4 | Photocatalytic H₂O₂ production and performance comparison.**
**a** Photocatalytic activity of TTI-COF and TTT-COF for H₂O₂ production (5 mg of COFs in 50 mL of water/BA (9:1) at 25 °C and λ > 420 nm). **b** H₂O₂ generation rates of photocatalytic reaction of TTT-COF treated with varying sulfur dosage. **c** Photocatalytic H₂O₂ production under different gas atmospheres. **d** Wavelength-dependent AQY measurement for TTT-COF. **e** Long-term photocatalytic H₂O₂ production of TTT-COF (inset is H₂O₂ production yield per day over an eight-day period). **f** Recyclability test of TTT-COF and TTI-COF. **g** Comparison of H₂O₂ generation rates and AQY of different photocatalysts. **h** PXRD patterns of TTI-COF and TTT-COF before and after four photocatalytic cycles (COFs were regenerated by washing with acetone and methanol). Experiments were repeated independently (**a**–**f**, **h**) three times with similar results. In **d**, **h**, a.u. indicates the arbitrary units. Source data are provided as a Source Data file.

capture of active intermediates, electron paramagnetic resonance (EPR) spectroscopy, in situ diffuse reflectance infrared Fourier transform spectroscopy (DRIFTS), and isotope labeling measurements. The introduction of p-benzoquinone (BQ, a scavenger of •O₂⁻) significantly suppressed H₂O₂ production, while the addition of t-butyl alcohol (TBA), a scavenger of •OH, had no effect on H₂O₂ generation (Fig. 5a). These findings indicate that H₂O₂ generation in this reaction system is predominantly dependent on electron transfer and necessitates the involvement of •O₂⁻ rather than •OH. Consequently, the

photoproduction of H₂O₂ is primarily associated with a two-step single-electron oxygen reduction reaction (ORR) process: $O_2 + e^- \rightarrow \cdot O_2^-$, $\cdot O_2^- + 2H^+ + e^- \rightarrow H_2O_2$. Additionally, when the holes were trapped in the presence of methanol (MeOH) and O₂, the H₂O₂ production for TTT-COF presented an upward trend. This phenomenon indicates that holes generated from TTT-COF may not be directly involved in the photocatalytic production of H₂O₂. Meanwhile, the H₂O₂ concentration was almost undetectable for TTT-COF when the electron-trapping agent (AgNO₃) was added in the presence of N₂. This result implies that

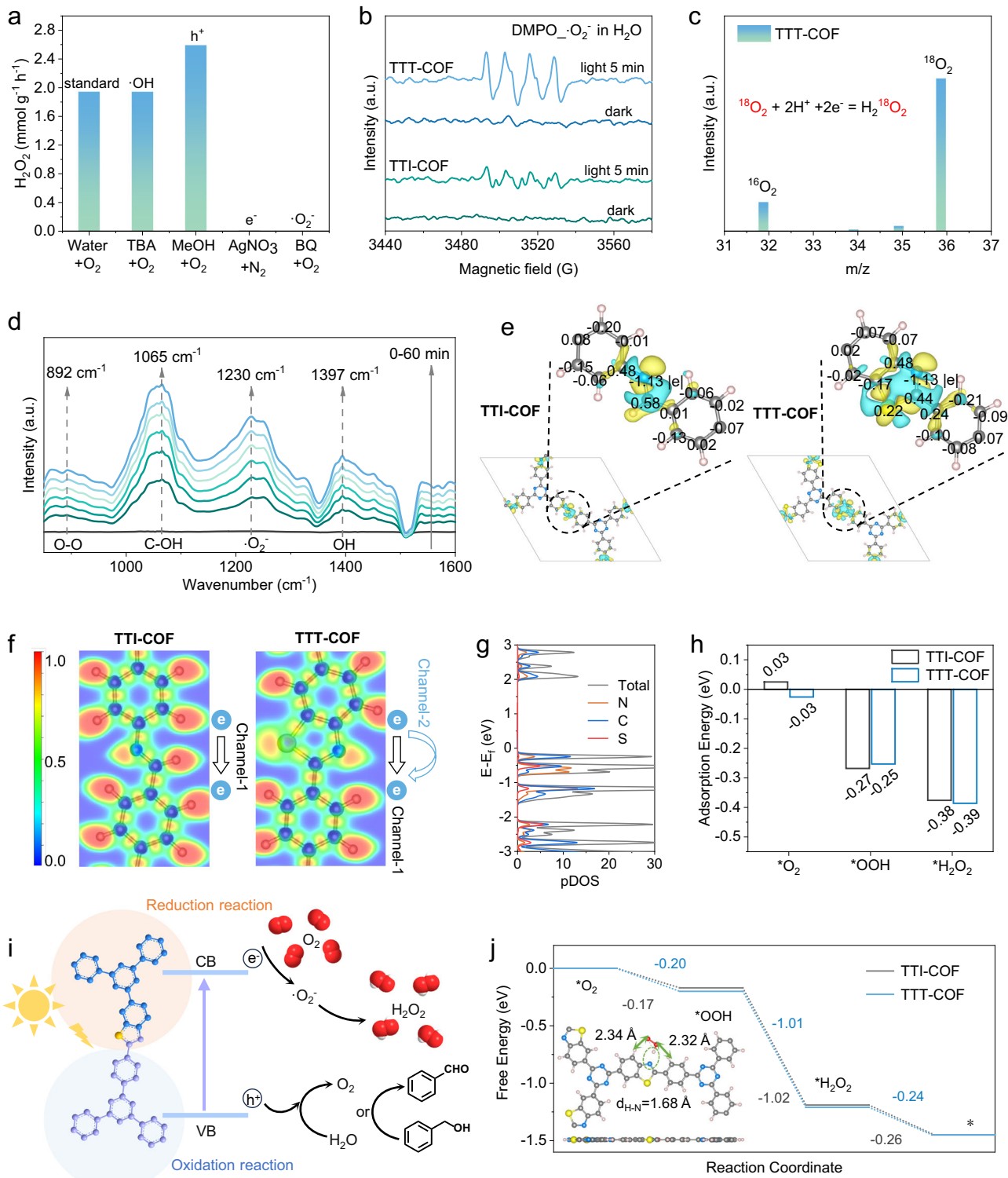

**Fig. 5 | Reaction pathways and mechanisms of $H_2O_2$ photosynthesis.**
**a** Photocatalytic $H_2O_2$ production for TTT-COF in neat water and water with TBA, MeOH, $AgNO_3$, and BQ (50 mL 10 mM aqueous solution, 25 mg COF), all with 1 h illumination. **b** EPR signals of the reaction solution under dark and visible light illumination in the presence of DMPO as the spin-trapping reagent. **c** $^{18}O_2$ isotope labeling experiment for TTT-COF to explore the source of $H_2O_2$. **d** In-situ DRIFT spectra of TTT-COF. **e** The charge density differences and Bader analysis of TTI-COF and TTT-COF. **f** The calculated ELF diagrams of TTI-COF and TTT-COF. **g** The computed partial density of states (pDOS) of TTT-COF. **h** The adsorption energy of TTI-COF and TTT-COF with oxygen-containing intermediates. **i** Schematic mechanism on photocatalytic $H_2O_2$ production of TTT-COF. **j** The calculated energy profiles for the reduction of $O_2$ into $H_2O_2$ of TTI-COF and TTT-COF. Experiments were repeated independently (**a**–**d**) three times with similar results. In **b**–**d**, a.u. indicates the arbitrary units. Source data are provided as a Source Data file.

a four-electron water oxidation reaction (WOR) may have occurred in TTT-COF ($2H_2O + 4h^+ \rightarrow O_2 + 4H^+$).

To confirm the presence of $\cdot O_2^-$ in the ORR pathway, EPR measurements were performed in methanol/$H_2O$ using 5,5-dimethyl-1-pyrroline N-oxide (DMPO) as a free-radical spin-trap agent. As depicted in Fig. 5b, upon visible light irradiation, the EPR spectra of both TTI-COF and TTT-COF reaction systems exhibited characteristic signals indicative of $\cdot O_2^-$. In contrast, no signals were detected in the absence of light, confirming the formation of $\cdot O_2^-$ within the photocatalytic system. Meanwhile, using 2,2,6,6-tetramethyl-4-piperidone (TEMP) as a spin-trapping agent of $^1O_2$, the EPR spectra of TTT-COF reaction systems exhibited a weak signal indicative of $^1O_2$ under visible light irradiation. In contrast, no signal was detected in TTI-COF, confirming that imine linkages with discontinuous π-conjugation impedes EnT process (Supplementary Fig. 43). However, the presence of $\cdot OH$ was not detected in the EPR measurements, which rules out the possibility of the $1e^-$ WOR process during the $H_2O_2$ photosynthesis in TTT-COF (Supplementary Fig. 44). In addition, isotope labeling experiments were performed using $^{18}O_2$ and $H_2^{18}O$, as shown in Fig. 5c and Supplementary Fig. 45. In the sealed photocatalytic reaction system with $^{18}O$-labeled oxygen molecules, the content of $^{18}O_2$ and $^{16}O_2$ was detected via decomposition of formed $H_2O_2$, a higher ratio of $^{18}O_2/^{16}O_2$ for TTT-COF was observed, indicating that $H_2^{18}O_2$ was the dominant product that came from the reduction of $^{18}O_2$ (Fig. 5c). Furthermore, when the photocatalytic reaction was conducted in pure $H_2^{18}O$, $^{18}O_2$ was detected directly from the headspace after photoirradiation, confirming the $4e^-$ WOR process in TTT-COF. Thereafter, we extracted the aqueous phase and then injected it into a vial containing $MnO_2$. $^{18}O_2$ was detected directly from the gas chromatography-mass spectrometer (GC-MS), which revealed that the generated $O_2$ from WOR could subsequently participate in the ORR to form $H_2O_2$ (Supplementary Fig. 45). Additionally, oxygen adsorption and desorption curves display that the adsorption capacity of $O_2$ on TTT-COF is significantly higher than that of TTI-COF, indicating that the thiazole unit in TTT-COF promotes $O_2$ adsorption (Supplementary Fig. 46).

DRIFTS was performed in-situ to identify the intermediates and photocatalytic active centers. The intensities of the characteristic peaks of $\cdot O_2^-$ ($1230\ cm^{-1}$) and $O-O$ ($892\ cm^{-1}$) increased progressively during the reaction in the in-situ DRIFTS tests for TTT-COF (Fig. 5d), signifying the adsorption of $O_2$ and the occurrence of the two-step single-electron pathway in the photocatalytic system based on TTT-COF. These comprehensive analyses reveal the pivotal role of $\cdot O_2^-$ in the photosynthesis of $H_2O_2$ and underscore the significance of the two-step single-electron ORR process in the overall mechanism. Meanwhile, signal peaks attributed to $H_2O$ oxidation intermediate C-OH and OH were observed at $1065\ cm^{-1}$ and $1397\ cm^{-1}$ for TTT-COF, proving the formation of adsorbed *OH for the WOR process. The insights gained from these experiments provide a robust foundation for further optimizing the photocatalytic performance of COFs in $H_2O_2$ generation and other related applications.

To investigate the origin of the catalytic activity of TTT-COF, density functional theory (DFT) calculations were conducted (Supplementary data 1)[80,81]. The total charge numbers of different components in TTI-COF and TTT-COF were first calculated, yielding 1.39 |e| for TTI-COF and 1.79 |e| for TTT-COF, respectively, to evaluate their intramolecular polarities. Such enhanced intramolecular polarity might boost the electron donor-acceptor effect and facilitate the charge separation in TTT-COF (Supplementary Fig. 47). Moreover, the electron localization functions (ELF) diagrams and the charge density differences of TTI-COF and TTT-COF were illustrated (Fig. 5e, f and Supplementary Fig. 48). Specifically, introducing the S atom altered the charge state of adjacent C/H moieties and extended the delocalized pathways for π electrons, leading to the superior intrinsic activity of TTT-COF. As shown in Supplementary Fig. 49, the orbital distribution of the conduction band minimum (CBM) and valence band

maximum (VBM) indicates that TTT-COF exhibits greater spatial separation characteristics, which partially suppresses electron-hole recombination, prolongs electron/hole lifetimes, and facilitates the progress of photocatalytic reactions. Furthermore, compared to TTI-COF, the doping of S atoms introduces additional electronic states near the Fermi level (Fig. 5g and Supplementary Fig. 50), which induce a redistribution of electronic configurations in adjacent C/N atoms, thereby enhancing intramolecular polarity.

Subsequently, we investigated the interaction between various oxygen-containing intermediates involved during the ORR process for both TTI-COF and TTT-COF, respectively (Supplementary Figs. 51, 52). TTT-COF showed a much better capturing capability of $O_2$ molecules than TTI-COF at the initial stage of the reaction (Fig. 5h). On the other hand, the formation of thiazole-linkage enabled homologous hetero-polyaromatic TTT-COF to maintain high conjugation in the molecular plane while capturing oxygen-containing intermediates, which is not the case for TTI-COF. This disruption of conjugation within the molecular plane inevitably led to the low efficiency of TTI-COF in the photocatalytic process. As shown in Fig. 5i, the oxygen reduction reaction occurs at the electron acceptor section (the thiazole unit), and water oxidation or benzyl alcohol oxidation occurs at the electron donor section. As depicted in Fig. 5j, which illustrates the free energy profile for the specific process of photocatalytic reduction of $O_2$ to $H_2O_2$, both TTI-COF and TTT-COF exhibit good photocatalytic properties; notably, the TTT-COF shows a particularly stronger intrinsic activity (-0.17 vs -0.20 eV). Therefore, TTT-COF demonstrates better photocatalytic activities, which is consistent with the above experimental observations.

## Photocatalytic performance toward aerobic oxidation of C(sp³)−H bonds

The chemical stability, efficient electron transfer, and prolonged charge carrier lifetimes of thiazole-based COFs position them as promising candidates for photocatalytic aerobic oxidation. We evaluated their performance using ethylbenzene oxidation as a model reaction, revealing striking differences between the fully cyclized TTT-COF and its TTI-COF counterpart. While TTT-COF achieved a high acetophenone yield (99%), TTI-COF showed no catalytic activity under identical conditions. Notably, partially cyclized TTT-COF (0.25eq) demonstrated intermediate performance (30% conversion), underscoring the critical role of complete thiazole ring formation in photocatalytic efficiency (Fig. 6b). Control experiments systematically confirmed the essential components of this photocatalytic system: the thiazole-incorporated COF structure, molecular oxygen as oxidant, and appropriate light irradiation all proved indispensable for reaction progression (Supplementary Table 8). These findings highlight how precise structural control over heteroaromatic connectivity in COFs can dramatically influence their photocatalytic performance in aerobic oxidation reactions.

To assess the generality of this photocatalytic system, we examined the oxidation of various ethylbenzene derivatives under standardized conditions (Fig. 6a). TTT-COF maintained efficient catalytic efficiency for substrates bearing electron-donating halogen substituents (p-F, p-Cl, p-Br), while the electron-withdrawing p-OCH₃ variant showed slightly reduced activity (86% yield). Notably, the framework demonstrated remarkable versatility in converting p-diethylbenzene to the industrially valuable p-diacetylbenzene with 98% yield—a challenging transformation for conventional catalysts. The structural robustness of TTT-COF was evidenced through five consecutive catalytic cycles without significant activity loss (Fig. 6c), with post-reaction PXRD and FTIR analyses confirming complete retention of crystallinity and chemical functionality (Supplementary Figs. 53, 54). This combination of broad substrate scope and high stability positions thiazole-based COFs as practical candidates for sustainable oxidation catalysis.

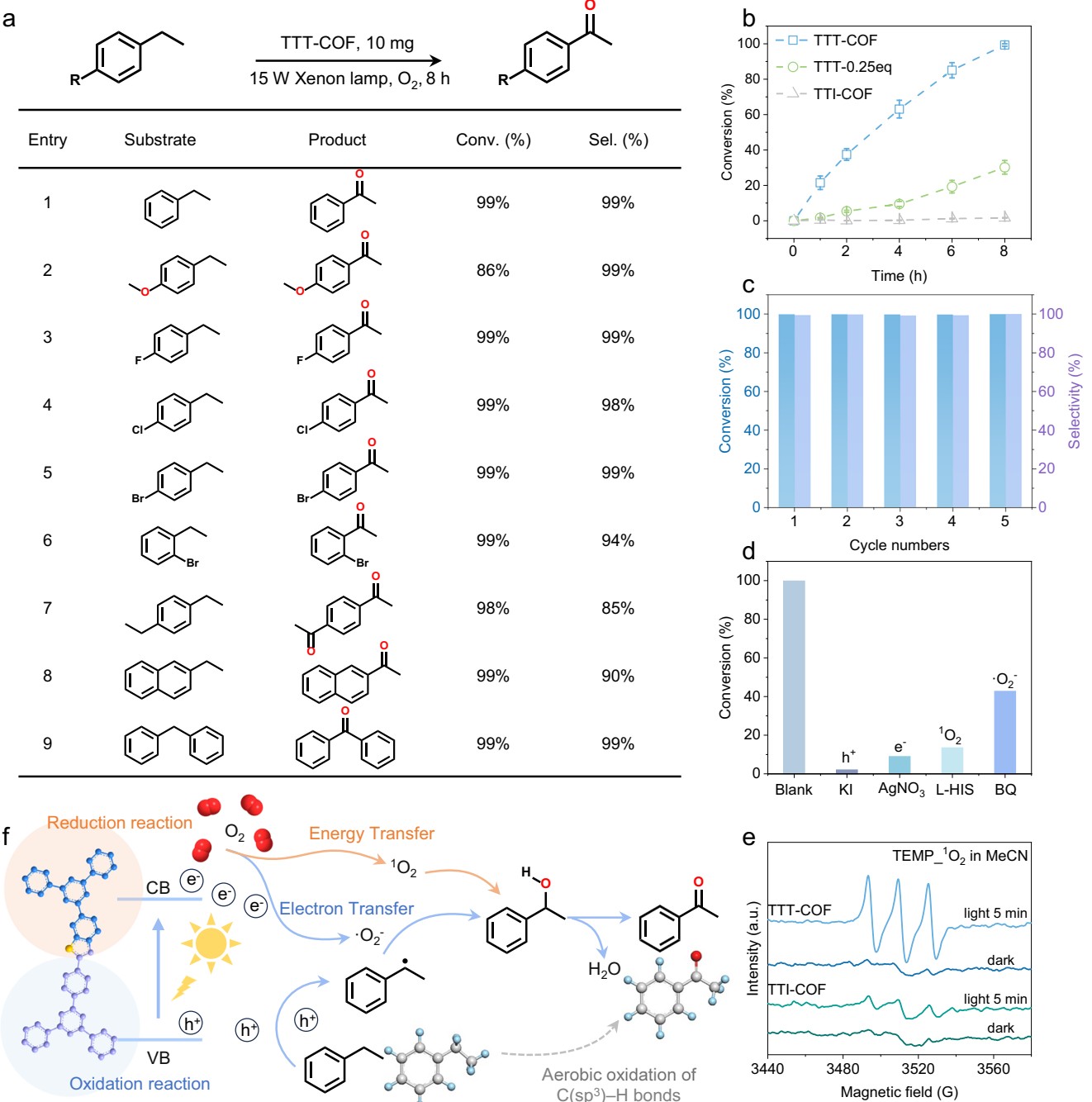

**Fig. 6 | The photocatalytic performance of TTT-COF for aerobic oxidation of C(sp³)-H bonds. a** TTT-COF catalyzed the aerobic oxidation of ethylbenzene derivatives. **b** Kinetic profile for oxidation of ethylbenzene. Reaction conditions: photocatalyst (10 mg), ethylbenzene (0.1 mmol), xenon lamp (420–1100 nm, 15 W), $CH_3CN$ (3 mL), $O_2$ (1 atm), 8 h. Determined by gas chromatography-flame ionization detector (GC–FID) using chlorobenzene as the internal standard, conversion of ethylbenzene, selectivity of acetophenone. Error bars indicate the error in the measurement. **c** Cycle experiment of aerobic oxidation of ethylbenzene along with TTT-COF as photocatalyst. **d** Control experiments for photocatalytic ethylbenzene oxidation by TTT-COF under normal conditions or with different scavengers. **e** EPR signals of the reaction solution under dark and visible light illumination in the presence of TEMP as the spin-trapping reagent. **f** Proposed mechanism for aerobic oxidation of C(sp³)-H bonds. Experiments were repeated independently (**a**–**e**) three times with similar results. In **e**, a.u. indicates the arbitrary units. Source data are provided as a Source Data file.

In order to explore the catalytic mechanism, we first identified the active species in the reaction process. A series of quenching experiments were performed by adding quenching agent to the original test solution of ethylbenzene oxidation (Fig. 6d). When a hole quencher (KI) or electron quencher ($AgNO_3$) was added, only a trace of the products could be detected after 8 h light irradiation, suggesting that photogenerated electrons and hole radicals are engaged in the reaction. When

a $^1O_2$ radical quencher (L-Histidine, L-HIS) and $\cdot O_2^-$ radical quenching agent (BQ) were introduced, the catalytic efficiency was significantly weakened, stating $O_2$ is reduced to $\cdot O_2^-$ and $^1O$, and involved in the oxidation reaction. To verify the generation of the above-mentioned ROSs, EPR spectroscopy in $CH_3CN$ (MeCN) of TTT-COF was performed. As shown in Fig. 6e and Supplementary Fig. 55, $\cdot O_2^-$ and $^1O_2$ signals appeared in the system after 5 min of illumination, indicating that TTT-

COF could rapidly activate oxygen molecules to ROS under light irradiation. However, no $^1O_2$ was detected in TTI-COF, which may be the main reason why TTI-COF cannot catalyze ethylbenzene oxidation. According to the obtained results, the reaction mechanism for the photocatalytic aerobic oxidation of $C(sp^3)$–H bonds is proposed in Fig. 6f. The presence of EnT in TTT-COF is attributed to the formation of homologous heteropolyaromatic structure with thiazole rings, which augment the overall conjugation of the framework and facilitates the formation of $^1O_2$, thereby facilitating efficient photocatalytic performance toward aerobic oxidation of $C(sp^3)$–H bonds.

## Discussion

In this work, by utilizing the sublimated sulfur vapor-induced post-cyclization reaction, we synthesize the thiazole-based homologous heteropolyaromatic TTT-COFs for enhancing photocatalytic $H_2O_2$ production and aerobic oxidation of $C(sp^3)$-H bonds. In general, this TTT-COF demonstrates three unique structural advantages: 1) the S heteroatoms in heteropolyaromatic COF structure enable remarkable chemical stability in harsh environments; 2) the unique structure with continuous π-conjugation of TTT-COF guarantees efficient SET and EnT for photocatalysis; 3) the thiazole-based frameworks can significantly enhance the intramolecular polarity, D-A structure, and charge separation, thus boosting the intrinsic photocatalytic activities and stabilities. Consequently, when compared to the original imine-linked COF structure, this TTT-COF exhibits a photosynthetic $H_2O_2$ production rate of 29.9 mmol $g^{-1}$ $h^{-1}$, which is a 20-fold improvement over the imine-linked COF and is competitive with many of the state-of-the-art organic and inorganic photocatalysts, including the other recently reported COFs. Meanwhile, its solar-to-chemical conversion efficiency for $H_2O_2$ production reaches 0.32%, exceeding the global average of natural photosynthetic organisms (-0.10%). Notably, the unique structure of TTT-COF enables it to have long-term photocatalytic stability that can operate continuously for more than 200 hours without a significant decline in activity. These findings offer perspectives on the design and development of chemically stable COFs and highlight their practical applications in photocatalytic reactions.

More importantly, benefited from the continuous π-conjugation structure, the constructed photocatalyst TTT-COF can undergo rapid intermolecular migration of photogenerated charges and substrate $(C(sp^3)$-H bonds) activation under light irradiation, so as to achieve high reactivity for oxidation of ethylbenzene with a conversion efficiency of 99.9% in 8 h, which is much higher than imine-based COF. In summary, the proposed strategy in this work paves the way to synthesize the next-generation homologous heteropolyaromatic COFs via a solvent-free, eco-friendly, and universal path and also provides insights to develop COF materials with tailored properties in large-scale production for a range of applications in photocatalysis, electrocatalysis, biomedical implants, and many other fields.

## Methods

### Synthesis of TTI-COF

A precise amount of tris(4-formylphenyl) triazine (TFPT) (19.7 mg, 0.05 mmol) and tris(4-aminophenyl) triazine (TAPT) (17.7 mg, 0.05 mmol) was combined with 1,4-dioxane (0.5 mL), mesitylene (0.5 mL), and acetic acid (6 M, 0.1 mL) in a Pyrex tube. The mixture was sonicated for 5 minutes to ensure homogeneity. Following this, the reaction mixture underwent degassing via a freeze–pump–thaw procedure for three cycles prior to being flame-sealed under vacuum conditions. Once the mixture reached room temperature (25 °C ± 2 °C), it was subjected to a thermal treatment at 120 °C for a duration of 3 days. The product was subsequently washed with tetrahydrofuran and acetone, and the resulting precipitate was extracted with tetrahydrofuran for 1 day using a Soxhlet extractor. The final step involved evaporating the solvent at 120 °C in a vacuum chamber overnight to yield the TTI-COF as a precipitate (32.7 mg, 94% yield).

### Synthesis of TTT-COF

The activated imine-linked TTI-COF (100 mg) was intimately mixed with molecular sulfur (1.5 g) using a mortar and pestle. After grinding for 5 minutes, the homogeneous mixture was transferred to a Pyrex tube, evacuated under vacuum, and purged with argon gas in a cycle repeated three times. The sealed tube was then placed in a furnace, and the temperature was ramped up to 155 °C at a rate of 2 °C per minute and held at this temperature for 3 hours. The temperature was further increased to 350 °C, maintaining the same heating rate, and kept at this elevated temperature for an additional 3 hours. Upon natural cooling, the brown-colored product was extracted using a Soxhlet apparatus with toluene and tetrahydrofuran for 12 hours each. The final product, COF powder, was washed with methanol and dried under vacuum at room temperature (25 ± 2 °C) to obtain the TTT-COF (90 mg, 90% yield). The TTT-xeq materials were prepared using the same procedure, referring to different sulfur dosages mixed with TTI-COF at mass ratios of 0.05, 0.25, 0.5, 1, 5, 10, and 15 using a mortar and pestle.

### Synthesis of TPI-COF

19.7 mg of tris(4-formylphenyl) triazine (TFPT) (0.05 mmol), 8.1 mg of p-phenylenediamine (PDA) (0.075 mmol), 1 mL of 1,4-dioxane, 1 mL of mesitylene, and 0.1 mL of 6 M acetic acid were discreetly mustered into a Pyrex tube and sonicated for 5 min. After that, the mixture was degassed via freeze–pump–thaw for three cycles and flame-sealed under a vacuum. Subsequently, the mixture was heated at 120 °C for 3 days and washed successively with DMF, THF, acetone, ethanol, and methanol. Lastly, after extracting with tetrahydrofuran for 1 day in a Soxhlet extractor, the remaining precipitate was totally evaporated at 120 °C in a vacuum chamber overnight (25.5 mg, 95% yield)[82].

### Synthesis of TPT-COF

An activated imine-linked TPI-COF (100 mg) was mixed with the molecular sulfur (1.5 g) in a mortar and pestle. After grinding for 5 minutes, the resulting homogeneous mixture was transferred to a Pyrex tube, evacuated under vacuum, and purged with argon three times. The entire mixture was further sealed under a vacuum and transferred to a furnace. The temperature was increased to 155 °C with a heating rate of 2 °C per minute and maintained for 3 h. Subsequently, the temperature was raised to 350 °C with the same heating rate and kept for another 3 h. After cooling down naturally, the resulting brown-colored material was washed using Soxhlet extraction with toluene and THF for 12 h, respectively. Finally, the COF powder was washed with methanol and dried at room temperature (25 ± 2 °C) (92 mg, 92% yield).

### Synthesis of BBI-COF

A precise amount of 1,3,5-tris(4-formlphenyl) benzene (TFPB) (19.5 mg, 0.05 mmol) and 1,3,5-tris(4-aminophenyl) benzene (TAPB) (17.6 mg, 0.05 mmol) was combined with 1,4-dioxane (0.5 mL), mesitylene (0.5 mL), and acetic acid (6 M, 0.1 mL) in a Pyrex tube. The mixture was sonicated for 5 minutes to ensure homogeneity. Following this, the reaction mixture underwent degassing via a freeze–pump–thaw procedure for three cycles prior to being flame-sealed under vacuum conditions. Once the mixture reached room temperature (25°C ± 2°C), it was subjected to a thermal treatment at 120 °C for a duration of 3 days. The product was subsequently washed with tetrahydrofuran and acetone, and the resulting precipitate was extracted with tetrahydrofuran for 1 day using a Soxhlet extractor. The final step involved evaporating the solvent at 120 °C in a vacuum chamber overnight to yield the BBI-COF as a precipitate (33.2 mg, 92%).

### Synthesis of BBT-COF

The activated imine-linked BBI-COF (100 mg) was intimately mixed with molecular sulfur (1.5 g) using a mortar and pestle. After grinding

for 5 minutes, the homogeneous mixture was transferred to a Pyrex tube, evacuated under vacuum, and purged with argon gas in a cycle repeated three times. The sealed tube was then placed in a furnace, and the temperature was ramped up to 155 °C at a rate of 2 °C per minute and held at this temperature for 3 hours. The temperature was further increased to 350 °C, maintaining the same heating rate, and kept at this elevated temperature for an additional 3 hours. Upon natural cooling, the brown-colored product was extracted using a Soxhlet apparatus with toluene and tetrahydrofuran for 12 hours each. The final product, COF powder, was washed with methanol and dried under vacuum at room temperature (25 ± 2 °C) to obtain the BBT-COF (90 mg, 90%).

## H₂O₂ detection method (Iodometry)

The amount of $H_2O_2$ was analyzed by Iodometry. 1 mL of 0.1 mol•L$^{-1}$ potassium hydrogen phthalate ($C_8H_5KO_4$) aqueous solution and 1 mL of 0.4 mol•L$^{-1}$ potassium iodide (KI) aqueous solution were added to 0.2 mL of the solution that was taken out from the catalytic system; the mixture was then kept for 30 min. $H_2O_2$ molecules react with iodine ions (I$^-$) under acidic conditions to generate triiodide ions (I$^{3-}$), which have strong absorption near 350 nm. The absorbance of I$^{3-}$ at 350 nm is measured by an ultraviolet spectrophotometer (UV-6100S), and then the amount of $H_2O_2$ generated by the photocatalytic reaction can be calculated.

## Photocatalytic H₂O₂ production

The photocatalytic $H_2O_2$ production experiment was conducted in a 60 mL glass vial, into which COFs (5 mg) and ultrapure water (45 mL) were introduced, along with a sacrificial agent (5 mL of ethanol or benzyl alcohol). The suspension was magnetically stirred for 10 minutes in the dark to ensure homogeneity, followed by sparging with oxygen (flow rate: 0.1 L/min) for 20 minutes to saturate the solution. The reaction system was then irradiated using a xenon lamp source (CEL-HXF300-T3, Beijing Zhongjiao Jinyuan Technology Co., Ltd., λ > 420 nm, equipped with circulating water), while continuously bubbling $O_2$ into the vial. The $H_2O_2$ concentration was quantified using a UV-Vis spectrophotometer as detailed in the supplementary experimental procedures.

## Photocatalytic reaction for the oxidation of ethylbenzene

A 25 mL quartz reactor was charged with 3 mL of chromatographic-grade acetonitrile and 10 mg of catalyst. The system underwent 5-minute gas purging (high-purity $N_2/O_2 \geq 99.995\%$, flow rate: 0.1 L/min) to displace ambient air unless atmospheric conditions were specifically required. Following gas equilibration, 0.1 mmol substrate was rapidly injected under continuous gas flow. Photocatalytic reactions proceeded under irradiation from a 15 W Xenon lamp (λ = 420–1100 nm) with temperature maintained at 298 K via cooling water circulation.

By the time of the reaction, the reaction mixture was collected, followed by centrifugation and 0.22 μm syringe filtration to isolate catalyst particles. For recyclability assessments, photocatalysts were retrieved by centrifugation, washed thoroughly with acetonitrile, and reused. The products were determined by GC-FID (GC-2010 Pro AF) using chlorobenzene as internal standard, with conversion/selectivity calculated from chromatographic data. Analogous procedures were employed for ethylbenzene derivatives at identical concentrations.

## Photoelectrochemical measurements

Electrochemical characterizations—including Mott–Schottky analysis, photocurrent response, and EIS—were conducted using a Gamry Reference 600 workstation (USA) configured with a standard three-electrode cell. For working electrode fabrication, 10 mg of catalyst was ultrasonically dispersed in a 0.9 mL ethanol/0.1 mL Nafion mixture to form a homogeneous ink. Subsequently, 50 μL of this slurry was uniformly coated onto ITO glass (1 × 1 cm$^2$, loading: 0.5 mg/cm$^2$) and infrared-dried. The three-electrode configuration employed a platinum wire counter electrode and an Ag/AgCl reference electrode. Before measurements, the Ag/AgCl electrode underwent RHE calibration in $H_2$-saturated electrolyte using a Pt wire, with all ORR potentials subsequently referenced to RHE. All electrolytes tested were 0.5 M $Na_2SO_4$, which was freshly prepared (within two weeks) and stored in a sealed container at room temperature (25 ± 2 °C). The specific configuration scheme is dissolving 35.5 g $Na_2SO_4$ in 500 mL ultrapure water to yield 0.5 M $Na_2SO_4$. The solution resistance was found to be 16.9 ± 0.5 Ω. All measurements were automatically corrected for iR drop using real-time resistance compensation.

EIS analysis was performed under dark conditions (frequency: 100 kHz–0.01 Hz, bias: 0 V) to determine charge transfer resistance (Rct). Mott–Schottky measurements spanned –1 to 1 V at 500, 1000, and 1500 Hz. Photocurrent testing employed a 300 W Xenon lamp (λ > 420 nm; CEL-HXF300-T3, Beijing Zhongjiao Jinyuan Technology Co., Ltd) for electrode illumination. Photocurrent transients were recorded over five 20-s on/off cycles (initial voltage: 0 V, sampling: 0.1 s, sensitivity: $1 \times 10^{-5}$ A V$^{-1}$).

## The AQY and SCC measurement

The photocatalytic reaction was carried out in pure deionized water (50 mL) and with a catalyst (20 mg) in a photocatalytic reactor. After sonication and bubbling, the bottle was irradiated by a Xenon lamp at 420-800 nm (CEL-HXF300-T3, Beijing Zhongjiao Jinyuan Technology Co., Ltd).

The AQY was determined under the irradiation of a 300 W Xenon lamp at a certain wavelength (λ = 400 nm, 420 nm, 450 nm, 475 nm, and 500 nm), and the light intensity was measured by a CEL-NP2000-2(10)A with a photodiode sensor. The AQY was calculated using the following equation:

$$AQY = \frac{(\text{Number of produces} H_2O_2 \text{molecules}) \times 2}{\text{Number of incident photons}} \times 100 \quad (1)$$

$$AQY = \frac{(M_{H_2O_2} \times N_A \times h \times c) \times 2}{S \times P \times T \times \lambda} \times 100 \quad (2)$$

Where M is the yield of $H_2O_2$ (mol), $N_A$ is Avogadro's constant ($6.022 \times 10^{23}$ mol$^{-1}$), h is the Planck constant ($6.626 \times 10^{-34}$ Js), c is the speed of light ($3 \times 10^8$ ms$^{-1}$), S is the irradiation area (cm$^2$), P is the intensity of irradiation light (W·cm$^{-2}$), T is the photoreaction time (s), λ is the wavelength of the monochromatic light (m).

The SCC efficiency was calculated by the following equation:

$$SCC\ efficiency(\%) = \frac{\left[\Delta G\ \text{for} H_2O_2\ \text{generation}\left(J mol^{-1}\right)\right] \times \left[H_2O_2\ \text{formed(mol)}\right]}{\left[\text{Total input power(W)}\right] \times \left[\text{Reaction times(s)}\right]} \times 100 \quad (3)$$

where ΔG = 117 kJ mol$^{-1}$, the irradiated area is 12.57 cm$^2$, the total input energy was therefore 1.257 W.

## EPR measurements

DMPO was used as a spin-trapping reagent to detect •OH or •$O_2^-$. In particular, the catalysts (2 mg) were dispersed into water or a MeOH/water mixture (9/1 v/v, 500 μL) containing DMPO (0.1 mmol) with a Pyrex glass tube, which was sealed with a rubber septum cap. A Xenon lamp (λ > 420 nm) was used as the light source. The dispersion was purged with Ar or $O_2$ gas (flow rate: 0.1 L/min) for 5 min before light irradiation. TEMP was served as the spin-trapping agent for the detection of singlet oxygen ($^1O_2$). The catalysts (2 mg) were dispersed into water (500 μL) containing TEMP (0.1 mmol) with a Pyrex glass

tube, which was sealed with a rubber septum cap. A Xenon lamp ($\lambda > 420$ nm) was used as the light source. The dispersion was purged with $O_2$ gas for 5 min before light irradiation.

In the aerobic oxidation of ethylbenzene, the EPR test is similar by replacing water with an acetonitrile/ethylbenzene mixture (9/1 v/v, 500 μL).

### Isotope labeling experiments

The $^{18}O_2$ isotope was measured on a GC-MS (Clarus 690, PerkinElmer). The detailed processes of the isotope labeling experiments were according to the literature method[83–86].

1. Photocatalysts (5 mg) and pure water ($H_2^{16}O$, 2 mL) were added to a 10 mL glass reactor. The reaction mixture was deoxygenated 3 times and backfilled with highly pure $N_2$ (99.999%, flow rate: 0.1 L/min). 5 mL $^{18}O_2$ was injected into the reactor via syringe, and the reaction was performed for 4 h under the irradiation of a 300 W xenon lamp at ambient temperature. In a separate 5 mL reactor, 100 mg of $MnO_2$ was introduced, and Ar was purged to eliminate $O_2$. The light-induced $H_2O_2$ solution was then added to the reactor containing $MnO_2$ to generate $O_2$, which was eventually quantified using a mass spectrometer.

$$^{18}O_2 + 2H^+ + 2e^- = H_2^{18}O_2$$
$$2H_2^{16}O + 2h^+ = H_2^{16}O_2 + 2H^+$$
$$2H_2O_2 \xrightarrow{MnO_2} 2H_2O + O_2$$

2. Photocatalysts (5 mg) and $H_2^{18}O$ (98%, 2 mL) were put into a hermetic device mainly composed of a quartz tube and sealing components (the air was pumped away with a vacuum pump). After $O_2$ was bubbled into the suspension in the dark for 30 min (flow rate: 0.1 L/min), the suspension was stirred in the dark for 30 min to reach the absorption-desorption equilibrium. After 36 h of irradiation, the gas products in the headspace of the reaction vessel were analyzed by GC-MS. Meanwhile, the formed $H_2O_2$ was decomposed by $MnO_2$ under an Ar atmosphere in another reactor. The $O_2$ generated by the decomposition of photogenerated $H_2O_2$ was analyzed by GC-MS.

$$2H_2^{18}O + 4h^+ = {}^{18}O_2 + 4H^+$$

### In-situ DRIFTS spectroscopy

In-situ DRIFTS measurements were performed based on the literature method[84], using a Nicolet iS-50 Fourier-transform spectrometer equipped with a Harrick diffuse reflectance accessory. Each spectrum was recorded by averaging 256 scans at a spectral resolution of 4 cm$^{-1}$. The samples were held in a custom-made IR reaction chamber, which was specifically designed to examine highly scattering powder samples in the diffuse reflection mode. The chamber was sealed with two ZnSe windows.

## Data availability

All data are available in the main text or the Supplementary Information. Source data are provided with this paper.

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

## Acknowledgements

This work was financially sponsored by the National Key R&D Program of China (grant no. 2024YFE0201200 [C.C.]), the National Natural Science Foundation of China (grant no. 52173133 [C.C.], 52373148 [C.C.], 52525311 [C.C.], 82302224 [X.H.X.]), the Sichuan Science and Technology Program (Nos. 2024YFHZ0270 [X.H.X.], 2024YFHZ0349 [X.K.L.]), and the State Key Laboratory of Polymer Materials Engineering (Grant No. sklpme2024-2-14 [X.H.X.]). Dr. Mi Zhou thanks the support from the Opening Project of the Sichuan Provincial Engineering Research Center of Functional Development and Application of High Performance Special Textile Materials (Chengdu Textile College, Project Number: 2024FDAST-C08). We would like to thank Dr Wu, Peng & Dr. Yanying Wang of the Analytical & Testing Center, Sichuan University, for their assistance on steady/transient fluorescence, Dr. Yani Xie for her assistance on solid-state UV-Vis diffuse reflectance spectra, and Dr. Yuanming Zhai for his assistance on nuclear magnetic resonance (NMR) spectra. We would like to thank the Sichuan University Interdisciplinary Innovation Fund.

## Author contributions

J.N.Y., C.C., and X.K.L. proposed the idea and designed the experiments. J.N.Y., X.L., and H.Y. performed the experiments, characterization, and results analysis. J.Y. and H.Y. assisted with the figure production and experiment design. J.N.Y., C.C., and Z.Y.Z. design and conduct the theoretical calculation. J.N.Y., X.H.X., and C.C. wrote and edited the manuscript. X.H.X., X.K.L., S.L., M. Z., and C.C. supervised the whole project. All authors discussed the results and commented on the manuscript.

## Competing interests

The authors declare no competing interests.
