## [Transparent Peer Review file · Nature Communications]

Homologous heteropolyaromatic covalent organic frameworks for enhancing photocatalytic hydrogen peroxide production and aerobic oxidation

Corresponding Author: Professor Chong Cheng

Version 0:

Reviewer comments:

Reviewer #1

(Remarks to the Author)

Dear Editor,

Cheng et al reported the design of an efficient, universal, solvent-free, and eco-friendly strategy to synthesize the thiazole-based COFs (TTT-COF) via sublimated sulfur vapor-induced post-cyclization reaction for the photocatalytic H₂O₂ generation and bacterial disinfection. TTT-COF exhibits a superior photosynthetic H₂O₂ production rate of 29.9 mmol g⁻¹ h⁻¹ and long-term stability. However, the novelty of this manuscript is not very clear. The sulfur vapor strategy has already been reported (NATURE COMMUNICATIONS | (2018) 9:2600). Furthermore, there are several issues that need to be solved before considered for publication.

1. Some of the points are difficult to understand. For example, "This diversity, while offering a rich tapestry of possibilities, also presents a formidable challenge in developing COF-based photocatalytic materials".
2. The term "homologous polyaromatic COFs" needs an explanation before use.
3. The sulfur vapor-induced post-cyclization reaction has been reported before, is there any difference?
4. The pore size of TTT-COF is labeled as 2.1 nm, but the pore width from the N₂ isotherm is 1.9 nm. The authors need to explain this discrepancy.
5. What about the conversion ratios from imine bonds to the thiazole molecule during the post-cyclization reaction?

Reviewer #2

(Remarks to the Author)

The manuscript investigates the thiazole-based homologous heteropolyaromatic COFs (TTT-COF) for the efficient H₂O₂ photosynthesis and bacterial disinfection. Compared with the imine-linked COF (TTI-COF), TTT-COF exhibits superior chemical stability, continuous π -conjugation, excellent electron transfer, and enhanced charge separation, thus achieving remarkable H₂O₂ photosynthesis rate of 29.9 mmol g⁻¹ h⁻¹ and photo-stabilities more than 200 hours. Besides, the TTT-COF-based personal protection mask can achieve >99.99% bactericidal efficiency against airborne drug-resistant bacteria within 30 minutes. This work is interesting and well organized, but some points need to be clarified or corrected. Some special comments are listed as follows.

1. In the abstract section, it should be emphasized that the H₂O₂ photosynthesis rate of 29.9 mmol g⁻¹ h⁻¹ by TTT-COF is achieved under the conditions involving organic sacrificial reagents.
2. In Figure 1d, "long-lived or short-lived charge separation" should be revised to "long-lived or short-lived charge carriers".
3. The peaks in FT-IR and ¹³C NMR of two COFs have not been fully assigned.
4. Figure S12 exhibits that TTI-COF has better thermal stability up to ~550°C compared to TTT-COF, which should be discussed in the main text.
5. The isotopic experiment should be conducted to further confirm the ORR process on COFs.
6. The CBM and VBM of TTT-COF do not match the D-A structure of TTT-COF identified in Figure S32. In addition, it is suggested to conduct the excited state charge analysis to assign donor and acceptor unit in COFs.
7. The partial density of states (PDOS) of TTT-COF and TTI-COF do not change significantly.
8. To further confirm the adsorption capacity of O₂ on COFs, the O₂-TPD experiments of TTI-COF and TTT-COF are

suggested to be characterized.

9. The possible active sites in both adsorption energy and reaction pathway calculations for ORR process should be fully considered.

Reviewer #3

(Remarks to the Author)

In this work, the authors synthesized the thiazole-based heteropolyaromatic COFs (TTT-COF) via sublimated sulfur vapor-induced post-cyclization reaction for photocatalytic H₂O₂ production and bacterial disinfection. Their studies indicated that the introduced elemental S heteroatom enables the modified COF with enhanced photophysical properties. Thus, TTT-COF showed a photosynthetic H₂O₂ production. After being integrated into the polypropylene fiber, the TTT-COF-based masks can present bactericidal effect against airborne drug-resistant bacteria. This work may be considered for publication after addressing the following issues:

1. What is the conversion yield from TTI-COF to TTT-COF? Will there is a full conversion? If not, how to purify the obtained TTT-COF in order to make sure that it is lack of impurities?
2. The conditions and experimental setup for photocatalytic H₂O₂ production can be complicated and varied with different materials. Is there a fair comparison in Figure 4f? Such claim should be careful.
3. The authors present two applications: H₂O₂ production and bacterial disinfection. It seems that it is disconnected between these two applications.
4. As a material for personal use in bacterial disinfection, the general toxicity of TTT-COF to cells or human should be evaluated.

Reviewer #4

(Remarks to the Author)

Review Comments

The manuscript by Cheng and co-workers reports the synthesis of two homologous COFs for photocatalytic H₂O₂ generation and bacterial disinfection. The post-synthetic modification method was used to synthesize thiazole-based homologous heteropolyaromatic TTT-COF from its parent imine-linked TTI-COF via post-cyclization reaction. The resultant TTT-COF exhibited enhanced chemical stability, continuous π -conjugation, excellent electron transfer, and enhanced D–A structure and charge separation, compared to the TTI-COF. The H₂O₂ production rate of TTT-COF was 3.71 mmol g⁻¹ h⁻¹ in pure water under air and further improved to 29.9 mmol g⁻¹ h⁻¹ by utilizing a sacrificial electron donor mixture (9:1 v/v water/benzyl alcohol (BA)) and continuously purging with O₂. The photocatalytic performance of TTT-COF is much superior to TTI-COF, which showed a production rate of only 0.48 mmol g⁻¹ h⁻¹ in pure water and air and 1.4 mmol g⁻¹ h⁻¹ in the presence of sacrificial reagent. Additionally, TTT-COF can be integrated into the polypropylene fiber and personal protection mask, which displayed >99.99% bactericidal efficiency against airborne drug-resistant bacteria within 30 minutes. Though the authors carried out lots of experiments and characterizations, the core novelty of the manuscript is not enough, and the present analysis is insufficient to verify some of their claims. Thus, I cannot recommend its publication.

1. The COF synthesis is simple and has been reported previously, which largely limited the novelty of this work. TTI-COF produced by imine polycondensation and TTT-COF generated by post modification are quite straightforward. TTI-COF has been reported by many research groups (Processes 2023, 11(10), 2859, ACS Materials Lett. 2024, 6, 11, 5016–5022, Mater. Chem. Front., 2017, 1, 1354-1361, Chem. Eur. J. 2020, 26, 5583).
2. The proposed oxidative cyclization synthetic strategy for TTT-COF is reported by other groups (J. Am. Chem. Soc. 2018, 140, 29, 9099–9103, Nat Commun 2018, 9, 2600), which should be clearly cited and demonstrated in the introduction. However, the manuscript provided a vague statement and citation, which is misleading and harmful to the research community.
3. The method for synthesis of TTT-COF is not clear. Whether the oxidation reaction was conducted under the oxygen atmosphere or not is unknown. Why was the reaction temperature maintained at 155 °C for 3 h before increasing to 350 °C? What is the yield of TTT-COF?
4. Before and after modification, the FTIR and ¹³C ssNMR spectra seem very similar, which may be due to the weak signals. This is difficult for claiming the formation of thiazole-linked TTT-COF. The corresponding characterizations of model compounds should be provided for validation.
5. In Figure 3, TTT-COF exhibited stronger photocurrent signals and smaller charge transfer resistance than TTI-COF. Any reasons for this enhancement?
6. TTT-COF displayed pronounced photoluminescence quenching and longer fluorescence lifetime compared with TTI-COF. Why does increasing conjugation of COF lead to these variations?
7. Though the proposed TTT-COF exhibited improved photocatalytic H₂O₂ production rate compared to the imine-linked TTI-COF, the overall photocatalytic performance of TTT-COF in either pure water or water/BA mixture is quite average in comparison to other reported photocatalytic COFs (Angew. Chem. Int. Ed. 2023, 62, e202218868, J. Am. Chem. Soc. 2024, 146, 20107–20115, Angew. Chem. Int. Ed. 2025, 64, e202416350).
8. The authors confirmed the superoxide anion radical as possible intermediate, but the photoactivation of oxygen usually accompanied with the production of singlet oxygen. Whether this active oxygen species involved in the photocatalytic reaction is not studied.
9. The mechanism investigation is not completed. The photocatalytic H₂O₂ production usually involves two half-reactions of oxygen reduction and water oxidation. The photocatalytic production of H₂O₂ can also go through the ·OH-involved two-step WOR process (Angew. Chem. Int. Ed. 2022, 61, e202200413) and the direct oxidation of water to H₂O₂, but the author did not investigate these pathways, so the proposed reaction mechanism is doubtful.

10. The catalytic sites were not investigated. Which molecular unit was used for oxygen reduction reaction? Which structure was used for oxidation half-reaction? The author should provide more explanations on Figure 5i and 5j.

11. In Figure 5g and Supplementary Figure 35, the author suggested that the doping of S atoms can introduce active electrons near the Fermi level in TTT-COF. It is difficult to observe the difference from their PDOS. More detailed explanations should be provided.

12. The use of COFs for bacterial disinfection via photocatalytic generation of reactive oxygen species is not new (Angew. Chem. Int. Ed. 2022, 61, e202200413, Angew. Chem. Int. Ed. 2024, 63, e202318562).

13. In Figure 6, TTT-COF was integrated into the polypropylene fiber (PP) and mask, which contains three layers. The PP with COF layer was positioned at the second layer that protected by the other two PP layers and cannot access the light. That is to say, the design is invalid as photocatalysis cannot proceed.

Version 1:

Reviewer comments:

Reviewer #1

(Remarks to the Author)

The authors have addressed the issues, and the manuscript is suitable for publication as is.

Reviewer #2

(Remarks to the Author)

All the concerns I raised previously have been addressed satisfactorily. Therefore, I recommend the manuscript for publication.

Reviewer #3

(Remarks to the Author)

The revised manuscript is recommended for publication.

Reviewer #4

(Remarks to the Author)

The revised manuscript added lots of new results, which improved the quality of the work greatly and answered my concerns. Thus I suggest publications of this paper.

Point-by-point response to the detailed comments by reviewers of “Homologous Heteropolyaromatic COFs via Post-Cyclization Reaction for Superior Photocatalytic H₂O₂ Production and Aerobic Oxidation of C(sp³)-H bonds” with manuscript ID: NCOMMS-24-84506A.

REVIEWER COMMENTS

Reviewer #1 (Remarks to the Author):

“Cheng et al reported the design of an efficient, universal, solvent-free, and eco-friendly strategy to synthesize the thiazole-based COFs (TTT-COF) via sublimated sulfur vapor-induced post-cyclization reaction for the photocatalytic H₂O₂ generation and bacterial disinfection. TTT-COF exhibits a superior photosynthetic H₂O₂ production rate of 29.9 mmol g⁻¹ h⁻¹ and long-term stability. However, the novelty of this manuscript is not very clear. The sulfur vapor strategy has already been reported (NATURE COMMUNICATIONS | (2018) 9:2600). Furthermore, there are several issues that need to be solved before considered for publication.”

Response to the general comment:

We are grateful for the constructive feedback on our manuscript. Your comments and suggestions have been instrumental in enhancing the quality of our work. In response to your insights, we have thoroughly revised the manuscript to enhance the novelty of our work. All queries and concerns have been addressed in the revised manuscript and the accompanying supplementary material. Your guidance has been invaluable in refining our study, and we are confident that these revisions have substantially improved the paper. Once again, we extend our thanks for your insightful contributions.

For the novelty comments: yes, we agree with the reviewer that the sulfur vapor strategy used in this work was inspired by the *Nat. Commun.*, **2018**, 9, 2600. The novelty of this work is not designing a new type of COF. The main purpose of our work is to investigate the effects of the post-cyclization reaction via sublimated sulfur vapor strategy on the optical/optoelectronic properties and corresponding photocatalytic performances of these homologous heteropolyaromatic COFs. Our systematic studies demonstrate that the sublimated sulfur vapor-introduced elemental S heteroatom enables modified COF materials with remarkable chemical stability, continuous π -conjugation,

excellent electron and energy transfer, and enhanced D-A structure and charge separation, thus boosting the intrinsic photocatalytic activities and stabilities. Consequently, TTT-COF achieves a superior photosynthetic H₂O₂ production rate of 29.9 mmol g⁻¹ h⁻¹ with more than 200 hours of long-term stability when employing 10 % benzyl alcohol (V/V) as a sacrificial agent in an O₂-saturated atmosphere, which is a 20-fold improvement over the imine-linked COF and exceeds many of the state-of-the-art photocatalysts. Notably, the TTT-COF photocatalyst can undergo rapid intermolecular migration of photogenerated charge, so as to achieve high reactivity for the oxidation of ethylbenzene derivatives. We wish that these interesting new findings on the optical/optoelectronic properties and photocatalytic performances of these thiazole-based homologous heteropolyaromatic COFs (TTT-COF) via solvent-free and eco-friendly post-cyclization reaction will clarify the novelty of this manuscript.

Furthermore, in the revised manuscript, in order to enhance the strength and significance of the article, we have performed the isotopic labelling experiments in order to clearly investigate the two half reactions mechanism; meanwhile, we have added and evaluated the photocatalytic performances of COFs toward aerobic oxidation of C(sp³)-H bonds. To strengthen the mechanistic analysis reported in the work, we further performed more comprehensive experiments on active intermediates capture and electron paramagnetic resonance. With the above significant revision, we hope we have improved the novelty of this manuscript. If the reviewer has any additional comments, please let us know. We will further enhance this work to meet the requirements.

(1) Some of the points are difficult to understand. For example, "This diversity, while offering a rich tapestry of possibilities, also presents a formidable challenge in developing COF-based photocatalytic materials".

Response to comment:

Thank you for your valuable comments. We have modified this sentence to “While offering a rich tapestry of possibilities, this structure diversities of COF-based photocatalysts presents a formidable challenge in analysing the corresponding relationships between the chemical structures and

photocatalytic performances.” Thank you again for your valuable input, which has helped us to improve the quality of our work.

(2) The term “homologous polyaromatic COFs” needs an explanation before use.

Response to comment:

Thank you for your helpful suggestions and we agree that this term need to be further explained. The corresponding explanation has been added in the revised manuscript, as also shown below:

Page 3 in the revised manuscript: “In contrast, homologous polyaromatic COFs refer to a class of COFs constructed from polycyclic aromatic hydrocarbon building units/monomers and bonding modes that share structural homology (*J. Am. Chem. Soc.*, **2024**, 146, 17131–17139). Thus, it is suggested that the high degree of structural similarity and uniformity of homologous polyaromatic COFs will facilitate a more coherent analysis of the structure and function of their underlying photocatalytic mechanisms.”

(3) The sulfur vapor-induced post-cyclization reaction has been reported before, is there any difference?

Response to comment:

Thanks for your insightful and helpful comments. Yes, we have referred this previous work for the sulfur vapor-induced post-cyclization reaction, but we also made some changes based on the earlier study. The synthetic method in the previous work is as follows: The respective imine COF was activated under high vacuum at 150 °C and subsequently mixed with a 15-fold amount (by weight) of sulfur in a ball mill. The resulting homogeneous mixture was transferred to a quartz boat in a horizontal tubular furnace and purged at 60 °C under flowing argon. The temperature was increased to 155 °C (60 K h⁻¹ heating rate) and maintained there for 3 h. Subsequently, the temperature was raised to 350 °C (100 K h⁻¹ heating rate) and kept for 3 h. After cooling down, the resulting material was washed via Soxhlet extraction with toluene and THF for 24 h, respectively. The samples were dried at 70 °C in an oven and then at 150 °C under high vacuum.

The synthesis method in our work is as follows: The activated imine-linked TTI-COF (100 mg) was intimately mixed with molecular sulfur (1.5 g) using a mortar and pestle. After grinding for 5 minutes, the homogeneous mixture was transferred to a Pyrex tube, evacuated under vacuum, and purged with argon gas in a cycle repeated three times. The sealed tube was then placed in a furnace, and the temperature was ramped up to 155 °C at a rate of 2 °C per minute and held at this temperature for 3 hours. The temperature was further increased to 350 °C, maintaining the same heating rate, and kept at this elevated temperature for an additional 3 hours. Upon natural cooling, the brown-colored product was extracted using a Soxhlet apparatus with toluene and tetrahydrofuran for 12 hours each. The final product, COF powder, was washed with methanol and dried under vacuum at room temperature to obtain the TTT-COF (90 mg, 90%). The TTT-xeq materials were prepared using the same procedure, referring to different sulfur dosages mixed with TTI-COF at mass ratios of 0.05, 0.25, 0.5, 1, 5, 10, and 15 using a mortar and pestle.

The general synthetic method and principle are the same as earlier work, and the specific experimental difference is that we transferred the homogeneous mixture to a Pyrex tube instead of a quartz boat, and then the mixture was evacuated under vacuum, and purged with argon gas in a cycle repeated three times. With this treatment, S vapor does not leak and interacts better with the TTT-COF, greatly improving the imine-to-thiazole linkages conversion ratios. Meanwhile, in order to determine the optimal S content during the post-cyclization reaction, TTI-COF samples were subjected to treatments with varying sulfur dosage. In addition, the main purpose of our paper is to investigate the effect of the post-cyclization reaction on the optical properties and photocatalytic performance of COFs, which is different from the previous work. Through the characterization of optoelectronic properties, the performance test on photocatalytic H₂O₂ production, the test of photocatalytic aerobic oxidation and the calculation of DFT, the following conclusions are obtained: this TTT-COF demonstrates three unique structural advantages (Fig. 1): 1) introducing the S heteroatom to form heteropolyaromatic COF enables remarkable chemical stability; 2) the unique structure with continuous π -conjugation of TTT-COF guarantees excellent single electron transfer (SET) and energy transfer (EnT) for photocatalysis; 3) the thiazole-based frameworks can significantly enhance the intramolecular polarity, D-A structure, and charge separation, thus boosting the intrinsic photocatalytic activities.

(4) *The pore size of TTT-COF is labeled as 2.1 nm, but the pore width from the N₂ isotherm is 1.9 nm. The authors need to explain this discrepancy.*

Response to comment:

Thank you for your attentive review and valuable feedback. There are several possible reasons for this discrepancy. First, due to the presence of some residual solvent molecules or substrate molecules that may lead to smaller pore sizes under N₂ isotherm. However, these molecules do not have an obvious effect on TEM photography since the electron beam will damage these small molecules. Moreover, the BET model is not so perfectly adapted to the pore structure of COF, and the presence of chemical groups in COF affects the gas adsorption process. In fact, there are some discontinuous pores and non-crystallization areas in the COF, which may lead to smaller pore sizes. However, the perfect pore structure areas were selected for TEM analysis, and some inevitable areas with defects in the COF were not observed. Therefore, the pore size of TTT-COF is labeled as 2.1 nm, which is closer to the theoretical value of COF structures.

(5) *What about the conversion ratios from imine bonds to the thiazole molecule during the post-cyclization reaction?*

Response to comment:

Thank you for your insightful comments and helpful suggestions regarding the conversion ratios from imine bonds to the thiazole molecules in our manuscript. In order to investigate the conversion ratios of the imine linkages to the thiazole linkages, the TTT-xeq materials were prepared using the same procedure as TTI-COF, referring to different sulfur dosages mixed with TTI-COF at mass ratios of 0.05, 0.25, 0.5, 1, 5, 10, and 15 using a mortar and pestle. Then, the obtained different samples were characterized by elemental analysis (EA), Fourier-transform infrared spectroscopy (FTIR), and powder X-ray diffraction (PXRD). As can be seen from the FTIR, the imine vibration located at 1627 cm⁻¹ completely disappears when the sulfur dosage is 0.5 times the mass of the TTI-COF. Meanwhile, EA tests show the presence of sulfur with an elemental composition close to the composition that would be expected from the thiazole unit. Thus, we can conclude that the conversion

from imine bonds to the thiazole molecule has been nearly completed when the sulfur dosage is 0.5 times the mass of the TTI-COF. However, when the sulfur dosage increases, PXRD data show that the crystallinity of TTT-COF also increases significantly. Therefore, when considering both the crystallization and the complete conversion, we choose the 15 times sulfur dosage to the mass of the TTI-COF, which can guarantee both the full conversion of imine bonds and high crystallinity of TTT-COF.

In summary, through the exploration of synthesis conditions, we found that TTI-COF could be completely transformed into highly crystallized TTT-COF at 15 times the sulfur dosage to the mass of the TTI-COF. In contrast, its imine bonds can only be partially transformed when the sulfur dosage is lower than 0.5 times the mass of the TTI-COF. The corresponding details can be found in the revised manuscript and revised supplementary information, as also shown below:

Page 7 in the revised manuscript: “Furthermore, to optimize the cyclization efficiency of TTT-COF, TTI-COF samples were subjected to treatments with varying sulfur dosage. The experimental results show that the optimal sulfur dosage for cyclization is determined to be 15 times the mass of TTI-COF (Supplementary Figs. 10, 11).”

The synthesis method in our work is as follows: The activated imine-linked TTI-COF (100 mg) was intimately mixed with molecular sulfur (1.5 g) using a mortar and pestle. After grinding for 5 minutes, the homogeneous mixture was transferred to a Pyrex tube, evacuated under vacuum, and purged with argon gas in a cycle repeated three times. The sealed tube was then placed in a furnace, and the temperature was ramped up to 155 °C at a rate of 2 °C per minute and held at this temperature for 3 hours. The temperature was further increased to 350 °C, maintaining the same heating rate, and kept at this elevated temperature for an additional 3 hours. Upon natural cooling, the brown-colored product was extracted using a Soxhlet apparatus with toluene and tetrahydrofuran for 12 hours each. The final product, COF powder, was washed with methanol and dried under vacuum at room temperature to obtain the TTT-COF (90 mg, 90%). The TTT-xeq materials were prepared using the same procedure, referring to different sulfur dosages mixed with TTI-COF at mass ratios of 0.05, 0.25, 0.5, 1, 5, 10, 15 using a mortar and pestle.”

Supplementary Figure 10. Powder X-ray diffraction (PXRD) patterns of TTT-COF treated with varying sulfur dosage, a.u. indicates the arbitrary units. The crystallinity of TTT-COF exhibited a progressive enhancement with increasing sulfur dosage, demonstrating sulfur's role in modulating the crystallization behavior of the COF during the synthetic process.

Supplementary Figure 11. Fourier-transform infrared spectroscopy (FTIR) spectra of TTT-COF treated with varying sulfur dosage, a.u. indicates the arbitrary units. FTIR analysis demonstrated a partial to complete conversion from imine bonds to thiazole bonds across sulfur dosages ranging from

0.05 to 15 times the mass of TTI-COF. PXRD characterization confirmed that the optimal sulfur dosage (15 times the mass of TTI-COF) achieved balanced cyclization efficiency and crystallinity preservation.

Supplementary Table 5. Elemental analysis of different TTT-COF samples obtained from treating TTI-COF with varying sulfur dosages.

	C	N	H	S
Expected	68.95	16.08	2.70	12.27
TTT-0.05eq	76.51	17.06	3.58	2.85
TTT-0.25eq	71.76	14.32	3.13	10.79
TTT-0.5eq	68.54	15.87	2.87	12.72
TTT-15eq	68.17	15.46	2.73	13.64

Reviewer #2 (Remarks to the Author):

“The manuscript investigates the thiazole-based homologous heteropolyaromatic COFs (TTT-COF) for the efficient H₂O₂ photosynthesis and bacterial disinfection. Compared with the imine-linked COF (TTI-COF), TTT-COF exhibits superior chemical stability, continuous π -conjugation, excellent electron transfer, and enhanced charge separation, thus achieving remarkable H₂O₂ photosynthesis rate of 29.9 mmol g⁻¹ h⁻¹ and photo-stabilities more than 200 hours. Besides, the TTT-COF-based personal protection mask can achieve >99.99% bactericidal efficiency against airborne drug-resistant bacteria within 30 minutes. This work is interesting and well organized, but some points need to be clarified or corrected. Some special comments are listed as follows.”

Response to the general comment:

We sincerely appreciate your recognition of our photocatalytic investigations on thiazole-based homologous heteropolyaromatic COF. Based on your comments and the suggestions from other reviewers, we have conducted more systematic experiments and refined the content throughout the manuscript. All necessary data have been added to support our claims, and we have thoroughly addressed all questions and concerns in the revised manuscript and supplementary information. Therefore, we believe that the quality of this paper has been significantly enhanced. We hope you will agree with this assessment, and we thank you once again for your considerable efforts.

(1) *In the abstract section, it should be emphasized that the H₂O₂ photosynthesis rate of 29.9 mmol g⁻¹ h⁻¹ by TTT-COF is achieved under the conditions involving organic sacrificial reagents.*

Response to comment:

Thanks for your important and helpful comments on improving the quality of our manuscript. We have changed the abstract to “Consequently, TTT-COF achieves a superior photosynthetic H₂O₂ production rate of 29.9 mmol g⁻¹ h⁻¹ with more than 200 hours of long-term stability when employing 10 % benzyl alcohol (V/V) as a sacrificial agent.”

(2) *In Figure 1d, “long-lived or short-lived charge separation” should be revised to “long-lived or short-lived charge carriers”.*

Response to comment:

We sincerely appreciate your insightful and constructive comments aimed at enhancing the quality of our manuscript. Your careful examination is greatly appreciated, and the updated corrections are included in the revised manuscript, as also illustrated below:

Fig. 1 Design of homologous heteropolyaromatic COFs via post-cyclization reaction. **a** Synthesis of triphenyl triazine thiazole COF (TTT-COF) from triphenyl triazine imine COF (TTI-COF) by the post-cyclization reaction from sublimated sulfur vapor, the inserted images are the synthesized COFs. **b** The Bader charge coloring distribution of TTI-COF. **c** Bader charge coloring distribution of TTT-COF. **d** The schematic illustration of the structural advantages of homologous heteropolyaromatic COFs (TTT-COF).

3) The peaks in FT-IR and ^{13}C NMR of two COFs have not been fully assigned.

Response to comment:

We sincerely appreciate your valuable comments and constructive feedback, which have played an essential role in improving our manuscript. We have fully assigned the peaks in FT-IR and ^{13}C NMR of two COFs in the revised manuscript, as also outlined below:

Fig. 2 Chemical structure properties of photocatalytic COFs. **g** ^{13}C ssNMR data of TTI-COF and TTT-COF.

Supplementary Table 3. Assignment and notes for TTI-COF.

peak (cm^{-1})	assignment and notes for TTI-COF
1627	imine C=N stretching
1576	aromatic ring stretching vibration
1511	triazine C=N stretching
1412	aromatic ring stretching vibration
1364	C-H bending vibration

815	aromatic ring C-H bending vibration
-----	-------------------------------------

Supplementary Table 4. Assignment and notes for TTT-COF.

peak (cm ⁻¹)	assignment and notes for TTT-COF
1609	thiazole N=C vibration
1576	aromatic ring stretching vibration
1564	aromatic C=C stretching vibration from thiazole-inducing
1511	triazine C=N stretching
1408	aromatic ring stretching vibration
1360	C-H bending vibration
815	aromatic ring C-H bending vibration

(4) *Figure S12 exhibits that TTI-COF has better thermal stability up to ~550°C compared to TTT-COF, which should be discussed in the main text.*

Response to comment:

Thank you for your invaluable feedback and thoughtful comments on our manuscript. We were sorry for not discussing the thermal stability in the main text, and we also carefully checked the data again. Probably due to the incorrect tests or deviation of the instrument, we decided to re-conduct the TGA experiment three times by using a new instrument, and the final results show that the TTT-COF is stable until 600 °C, while the TTI-COF began to decompose at 550 °C, which means that the thermal stability of the COF has also been greatly improved after sublimated sulfur vapor-induced post-cyclization treatment. The corresponding details can be found in the revised manuscript and revised supplementary information, as shown below:

Page 7 in the revised manuscript: “Moreover, thermogravimetric analysis experiment exhibits that TTT-COF has better thermal stability up to ~600 °C compared to TTI-COF (Supplementary Fig. 14). This detailed characterization not only validates the successful synthesis and structural integrity of

homologous heteropolyaromatic TTT-COF but also highlights its remarkable chemical and thermal stability, a critical attribute for applications in harsh chemical environments and photocatalytic processes.

Supplementary Figure 14. Thermogravimetric analysis (TGA) of TTI-COF and TTT-COF under N₂ atmosphere.

(5) The isotopic experiment should be conducted to further confirm the ORR process on COFs.

Response to comment:

Thanks for your important and helpful comments on improving the quality of our manuscript. We acknowledge your suggestion to conduct the isotopic experiment to confirm the ORR process further. Indeed, before we carry out this isotopic experiment, to ensure our reliability of our experimental conditions, we have already reviewed the experimental details abundant earlier literature (*Nat. Commun.*, **2023**, 14, 5238; *J. Am. Chem. Soc.*, **2024**, 146, 29943-29954; *Angew. Chem. Int. Ed.*, **2024**, 63, e20241017; *Nat. Catal.*, **2024**, 7, 195-206). The final protocol is as follows:

Photocatalysts (5 mg) and pure water (H₂¹⁶O, 2 mL) were added to a 10 mL glass reactor. The reaction mixture was deoxygenated 3 times and backfilled with highly pure N₂ (99.999%). 5 mL ¹⁸O₂ was injected into the reactor via syringe, and the reaction was performed for 4 h under the irradiation of a 300 W xenon lamp at ambient temperature. In a separate 5 mL reactor, 100 mg of MnO₂ was introduced, and Ar was purged to eliminate O₂. The light-induced H₂O₂ solution was then added to the

reactor containing MnO₂ to generate O₂, which was eventually quantified using a mass spectrometer.

Eventually, the experimental results showed that a higher ratio of ¹⁸O₂ and ¹⁶O₂ for TTT-COF was observed, indicating that H₂¹⁸O₂ was the dominant product that came from the reduction of ¹⁸O₂. The corresponding details can be found in the revised manuscript and revised supplementary information, as also shown below:

Page 15 in the revised manuscript: “In addition, isotope labeling experiments were performed using ¹⁸O₂ and H₂¹⁸O, as shown in Fig. 5c and Supplementary Fig. 45. In the sealed photocatalytic reaction system with ¹⁸O-labeled oxygen molecules, the content of ¹⁸O₂ and ¹⁶O₂ was detected via decomposition of formed H₂O₂, a higher ratio of ¹⁸O₂/¹⁶O₂ for TTT-COF was observed, indicating that H₂¹⁸O₂ was the dominant product that came from the reduction of ¹⁸O₂ (Fig. 5c).”

Fig. 5 Reaction pathways and mechanisms of H₂O₂ photosynthesis. **c** ¹⁸O₂ isotope labeling experiment for TTT-COF to explore the source of H₂O₂.

(6) The CBM and VBM of TTT-COF do not match the D-A structure of TTT-COF identified in Figure S32. In addition, it is suggested to conduct the excited state charge analysis to assign donor and acceptor unit in COFs.

Response to comment:

We sincerely appreciate the reviewers' valuable comments, which are crucial for enhancing the quality of this work. We are sorry that the CBM and VBM of TTT-COF do not match the D-A structure of TTT-COF identified through charge distribution analysis. Typically, we agree that the D-A division should align with the VBM (commonly assigned to the donor, D) and CBM (generally attributed to the acceptor, A) (*Nat. Commun.* **2023**, 14 (1), 4344; *Nat. Commun.* **2023**, 14 (1), 5238). To address this discrepancy, we have systematically analyzed the data and reassigned the D-A components. As illustrated in Figure S47, the incorporation of thiazole-sulfur significantly enhances the intramolecular polarity (increasing from 1.39 to 1.79 |e|), thereby strengthening the D-A interaction in TTT-COF. Furthermore, we have supplemented the excited state charge analyses for both TTI-COF and TTT-COF. As shown in Supplementary Figure 49, the introduction of thiazole S significantly enhanced the intramolecular polarity (increasing from 1.39 to 1.79 |e|), enabling TTT-COF to have a stronger D-A effect. Furthermore, we supplemented the excitation charge distributions of TIT-COF and TTT-COF. As shown in Supplementary Fig. 49, the introduction of thiazole S significantly makes TTT-COF exhibit greater spatial separation characteristics, which partially suppresses electron-hole recombination, prolongs electron/hole lifetimes, and can also effectively act as the active site for the reduction of O₂ to H₂O₂, thereby facilitating photocatalytic reaction kinetics. The corresponding details can be found in the revised manuscript and revised supplementary information, as also shown below:

Page 16 in the revised manuscript: “Total charge numbers of different components in TTI-COF and TTT-COF were first calculated, 1.39 |e| for TTI-COF and 1.79 |e| for TTT-COF, respectively, to evaluate their intramolecular polarities.

As shown in Supplementary Fig. 49, the orbital distribution of the conduction band minimum (CBM) and valence band maximum (VBM) indicates that TTT-COF exhibits greater spatial separation characteristics, which partially suppresses electron-hole recombination, prolongs electron/hole lifetimes, and facilitates the progress of photocatalytic reactions.”

Supplementary Figure 47. The calculated charge distribution of (a) TTI-COF and (b) TTT-COF structures. (color code: orange, S; gray, C; sky blue, N; and white, H).

Supplementary Figure 49. a) The calculated conduction band minimum (CBM) and valence band maximum (VBM) diagrams of TTI-COF and TTT-COF. b) The excited state charge analysis of TTI-COF and TTT-COF.

(7) The partial density of states (PDOS) of TTT-COF and TTI-COF do not change significantly.

Response to comment:

We sincerely appreciate the reviewers' insightful comments, we have carefully revised this data. To enable systematic comparison, the pDOS of both TTI-COF and TTT-COF have been consolidated into a unified plot to evaluate the effects induced by S atom incorporation. As shown in Fig. 5g and Supplementary Fig. 50, the introduction of S atoms indeed generates new electronic states proximal to

the Fermi level. However, considering that S does not directly participate in the H_2O_2 production pathway but rather indirectly modulates the electronic distribution of adjacent C/N atoms to enhance intramolecular polarity, we have accordingly refined the relevant descriptions as follows:

Page 17 in the revised manuscript: “Furthermore, compared to TTI-COF, the doping of S atoms introduces additional electronic states near the Fermi level (Fig. 5g and Supplementary Fig. 50), which induce a redistribution of electronic configurations in adjacent C/N atoms, thereby enhancing intramolecular polarity.”

Supplementary Figure 50. The computed pDOS of TTT-COF and TTI-COF.

(8) *To further confirm the adsorption capacity of O_2 on COFs, the O_2 -TPD experiments of TTI-COF and TTT-COF are suggested to be characterized.*

Response to comment:

We sincerely appreciate your valuable comments and constructive feedback, which have played an essential role in improving our manuscript. We fully agree with the reviewer's suggestion to conduct O_2 -TPD experiments of TTI-COF and TTT-COF to further confirm the adsorption capacity of O_2 on COFs. According to your comments, we have incorporated O_2 -TPD experiments into the revised manuscript, as also outlined below:

Page 16 in the revised manuscript: “Additionally, oxygen adsorption and desorption curves display that the adsorption capacity of O₂ on TTT-COF is higher than that of TTI-COF, indicating that the thiazole unit in TTT-COF promotes O₂ adsorption (Supplementary Fig. 46).”

Supplementary Figure 46. The O₂-TPD curves of TTT-COF and TTI-COF, a.u. indicates the arbitrary units.

9) *The possible active sites in both adsorption energy and reaction pathway calculations for ORR process should be fully considered.*

Response to comment: We sincerely appreciate the reviewers' valuable comments. During the calculation of adsorption energy and reaction pathways for the ORR process, we have fully considered the possible active sites in COF. The configurations shown in the manuscript are all ground-state configurations. Taking the TTT-COF*OOH system as a representative example, the alternative possible adsorption configurations that have been calculated are systematically illustrated as follows:

Supplementary Figure 52. The optimized structure model of TTT-COF*OOH. (color code: yellow, S; gray, C; sky blue, N; red, O; and white, H).

Reviewer #3 (Remarks to the Author):

“In this work, the authors synthesized the thiazole-based heteropolyaromatic COFs (TTT-COF) via sublimated sulfur vapor-induced post-cyclization reaction for photocatalytic H₂O₂ production and bacterial disinfection. Their studies indicated that the introduced elemental S heteroatom enables the modified COF with enhanced photophysical properties. Thus, TTT-COF showed a photosynthetic H₂O₂ production. After being integrated into the polypropylene fiber, the TTT-COF-based masks can present bactericidal effect against airborne drug-resistant bacteria. This work may be considered for publication after addressing the following issues:”

Response to the general comment:

Thank you for your invaluable feedback and thoughtful comments on our manuscript. We greatly appreciate the insights you provided; these comments have played a crucial role in enhancing the quality of our work. In the revised manuscript, based on the editor's and other reviewers' suggestions, we have moved the bacterial disinfection section of the study to supporting information. Meanwhile, as suggested by the other reviewers, to enhance the strength and significance of the article, we have added and evaluated the photocatalytic performances of TTT-COFs toward aerobic oxidation of C(sp³)-H bonds, and we believe that the quality and novelty of the revised manuscript has been significantly improved with these helpful comments. Thank you once again for your generous support and the time you dedicated to reviewing this paper.

(1) What is the conversion yield from TTI-COF to TTT-COF? Will there is a full conversion? If not, how to purify the obtained TTT-COF in order to make sure that it is lack of impurities?

Response to comment:

Thank you for your valuable comments. In order to investigate the conversion ratios of the imine linkages to the thiazole linkages, the TTT-xeq materials were prepared using the same procedure as TTI-COF, referring to different sulfur dosages mixed with TTI-COF at mass ratios of 0.05, 0.25, 0.5, 1, 5, 10, and 15 using a mortar and pestle. Then, the obtained different samples were characterized by elemental analysis (EA), Fourier-transform infrared spectroscopy (FTIR), and powder X-ray diffraction (PXRD). As can be seen from the FTIR, the imine vibration located at 1627 cm⁻¹

completely disappears when the sulfur dosage is 0.5 times the mass of the TTI-COF. Meanwhile, EA tests show the presence of sulfur with an elemental composition close to the composition that would be expected from the thiazole unit. Thus, we can conclude that the conversion from imine bonds to the thiazole molecule has been nearly completed when the sulfur dosage is 0.5 times the mass of the TTI-COF. However, when the sulfur dosage increases, PXRD data show that the crystallinity of TTT-COF also increases significantly. Therefore, when considering both the crystallization and the complete conversion, we choose the 15 times sulfur dosage to the mass of the TTI-COF, which can guarantee both the full conversion of imine bonds and high crystallinity of TTT-COF.

In summary, through the exploration of synthesis conditions, we found that TTI-COF could be completely transformed into highly crystallized TTT-COF at 15 times the sulfur dosage to the mass of the TTI-COF. In contrast, its imine bonds can only be partially transformed when the sulfur dosage is lower than 0.5 times the mass of the TTI-COF. The corresponding details can be found in the revised manuscript and revised supplementary information, as also shown below:

Page 7 in the revised manuscript: “Furthermore, to optimize the cyclization efficiency of TTT-COF, TTI-COF samples were subjected to treatments with varying sulfur dosage. The experimental results show that the optimal sulfur dosage for cyclization is determined to be 15 times the mass of TTI-COF (Supplementary Figs. 10, 11).”

The synthesis method in our work is as follows: The activated imine-linked TTI-COF (100 mg) was intimately mixed with molecular sulfur (1.5 g) using a mortar and pestle. After grinding for 5 minutes, the homogeneous mixture was transferred to a Pyrex tube, evacuated under vacuum, and purged with argon gas in a cycle repeated three times. The sealed tube was then placed in a furnace, and the temperature was ramped up to 155 °C at a rate of 2 °C per minute and held at this temperature for 3 hours. The temperature was further increased to 350 °C, maintaining the same heating rate, and kept at this elevated temperature for an additional 3 hours. Upon natural cooling, the brown-colored product was extracted using a Soxhlet apparatus with toluene and tetrahydrofuran for 12 hours each. The final product, COF powder, was washed with methanol and dried under vacuum at room temperature to obtain the TTT-COF (90 mg, 90%). The TTT-xeq materials were prepared using the same procedure, referring to different sulfur dosages mixed with TTI-COF at mass ratios of 0.05, 0.25, 0.5, 1, 5, 10, 15 using a mortar and pestle.”

Supplementary Figure 10. Powder X-ray diffraction (PXRD) patterns of TTT-COF treated with varying sulfur dosage, a.u. indicates the arbitrary units. The crystallinity of TTT-COF exhibited a progressive enhancement with increasing sulfur dosage, demonstrating sulfur's role in modulating the crystallization behavior of the COF during the synthetic process.

Supplementary Figure 11. Fourier-transform infrared spectroscopy (FTIR) spectra of TTT-COF treated with varying sulfur dosage, a.u. indicates the arbitrary units. FTIR analysis demonstrated a partial to complete conversion from imine bonds to thiazole bonds across sulfur dosages ranging from

0.05 to 15 times the mass of TTI-COF. PXRD characterization confirmed that the optimal sulfur dosage (15 times the mass of TTI-COF) achieved balanced cyclization efficiency and crystallinity preservation.

Supplementary Table 5. Elemental analysis of different TTT-COF samples obtained from treating TTI-COF with varying sulfur dosages.

	C	N	H	S
Expected	68.95	16.08	2.70	12.27
TTT-0.05eq	76.51	17.06	3.58	2.85
TTT-0.25eq	71.76	14.32	3.13	10.79
TTT-0.5eq	68.54	15.87	2.87	12.72
TTT-15eq	68.17	15.46	2.73	13.64

(2) *The conditions and experimental setup for photocatalytic H₂O₂ production can be complicated and varied with different materials. Is there a fair comparison in Figure 4f? Such claim should be careful.*

Response to comment:

Thanks for your good comment and helpful suggestion. Indeed, we agree with the reviewer that the conditions and experimental setup for photocatalytic H₂O₂ production are complicated. The production of photocatalytic H₂O₂ is related to the type of sacrificial agent, the content of the sacrificial agent, and even the light source. Therefore, for a fair comparison, we have listed the reaction conditions of each catalyst in the revised supporting information, and unified their units of measurement as mmol g⁻¹ h⁻¹. Meanwhile, we use the most common testing methods for photocatalytic performance comparison with sacrificial agents in our work, and the performances of the mentioned photocatalysts in Figure 4g are also obtained with sacrificial agents. Therefore, after revision, we have made a reasonable comparison of different photocatalysts in Figure 4g. The corresponding details can be found in the revised supplementary information, as shown below:

Supplementary Table 7. Performance and AQY comparison of TTT-COF with other photocatalysts

reported in the literature for H₂O₂ production.

Photocatalysts	Solution	Irradiation conditions /nm	H ₂ O ₂ yields/ mmol g ⁻¹ h ⁻¹	AQY (%)	Ref.
TTT-COF	H ₂ O:BA(9:1)	λ>420 nm	29.905	12% at 400 nm	This work
	pure water	λ>420 nm	4.75	NT	
TTH-CTP	H ₂ O:BA(9:1)	300 W Xenon lamp	23.7	11.3% at 450 nm	2
Au@COF	H ₂ O:BA (9:1)	Simulated sunlight	18.933	NT	3
DBTP-COF	H ₂ O:IPA (9:1)	λ>420 nm	15	7.4% at 450 nm	4
PD ²⁺ -COF _{16.7}	H ₂ O:EtOH (9:1)	λ>400 nm	11.86	12.9 % at 400 nm	5
sonoCOF-F2	H ₂ O:BA (9:1)	λ>420 nm	2.422	4.8% at 420 nm	6
TAPT-TFPA COFs@Pd ICs	H ₂ O:EtOH (9:1)	AM 1.5G	2.143	6.5% at 400 nm	7
CTF-NS-5BT	H ₂ O:BA (9:1)	λ>420 nm	1.630	6.7% at 420 nm	8
BTC40	H ₂ O: EtOH (9:1)	200-620nm	3.749	NT	9
1H-COF	H ₂ O:IPA (9:1)	λ>420 nm	1.483	5.4% at 420 nm	10
TF ₅₀ -COF	H ₂ O:EtOH (9:1)	λ>400 nm	1.739	5.1% at 400 nm	11
COF-NUST-16	H ₂ O:EtOH (9:1)	λ>420 nm	1.081	NT	12
Py-Da-COF	H ₂ O:BA (9:1)	λ>420 nm	3.670	4.5% at 420 nm	13
COF-TAPB-BPDA	H ₂ O:BA (19:1)	λ>420 nm	1.240	NT	14
TAPD-(OMe) ₂	H ₂ O: EtOH (9:1)	420~700nm	0.091	NT	15
TTA-TTTA	H ₂ O: EtOH (9:1)	λ~420 nm	4.347	NT	16
CN-COF	H ₂ O: EtOH (9:1)	λ>400 nm	2.623	9.8% at 420nm	17
ZT-5	H ₂ O: EtOH (9:1)	λ≥360 nm	2.443	13.12 at 365nm	18
PMCR-1	H ₂ O:BA(10:1)	λ>420 nm	5.5	NT	19
DMCR-1NH	H ₂ O:IPA (10:1)	λ>420 nm	2.588	10.2 % at 420 nm	20
CTF-LTZ	H ₂ O: EtOH (9:1)	full spectrum	4.068	4.5% at 400nm	21
5Cv@g-C ₃ N ₄	H ₂ O:EtOH (9:1)	λ>420 nm	7.010	9.58% at 420 nm	22

ZnO/g-C ₃ N ₄	H ₂ O:EtOH (9:1)	$\lambda > 350$ nm	3.860	NT	23
ZnO/WO ₃	H ₂ O:EtOH (9:1)	$300 \leq \lambda \leq 700$	6.788	12.5% at 365 nm	24
OPA/Zr _{100-x} Ti _x -MOF	H ₂ O:BA (2:5)	$\lambda > 420$ nm	13.580	NT	25
TaptBtt	Pure water	AM 1.5G	1.407	4.6 % at 450 nm,	26
COF-N32	Pure water	$\lambda > 420$ nm	0.605	6.2% at 459 nm	27
COF-TfpBpy	Pure water	$\lambda > 420$ nm	1.037	6.7% at 420 nm	28
CTF-BDDBN	Pure water	AM 1.5G	0.887	NT	29
Pt/TiO ₂	Pure water	$\lambda > 300$ nm	5.096	NT	30
ZnIn ₂ S ₄ /TiO ₂	Pure water	$400 \text{ nm} \leq \lambda \leq 760$	1.530	NT	31
CNIO-GaSA	Pure water	$\lambda > 420$ nm	0.332	7.1% at 459 nm	32

BA = benzyl alcohol; EtOH = ethanol; IPA = isopropanol; NT: Not Tested

(3) The authors present two applications: H₂O₂ production and bacterial disinfection. It seems that it is disconnected between these two applications.

Thank you for your valuable comments; we agree with the reviewer that there is some disconnect between these two applications. In our earlier experimental design, the method of using H₂O₂ produced by photocatalysis for bacterial disinfection has also been utilized in some literatures (*Angew. Chem. Int. Ed.*, **2022**, 61, e202200413; *J. Am. Chem. Soc.*, **2024**, 146, 31950-31960). In a continuous photocatalytic production of H₂O₂ test, TTT-COF maintained a steady H₂O₂ production rate of ~4 mM h⁻¹ over an extended period of 200 hours, with no significant degradation in catalytic activity. By the end of this test, the H₂O₂ concentration in the solution reaches 0.735 M, a level that is considered practical for highly effective antimicrobial applications. Thus, the synthesized COF photocatalysts may show promising potential applications in future disinfection therapies.

However, considering the suggestions from the editor and other reviewers, we have decided to remove the bacterial disinfection applications to the supporting information in order to make the overall paper focus on the photocatalysis. Meanwhile, we have added the photocatalytic performance toward aerobic oxidation of C(sp³)-H bonds of COFs in order to enhance the quality and novelty of

this work. The results show that the constructed photocatalyst TTT-COF can undergo rapid intermolecular migration of photogenerated charges and substrate (C(sp³)-H bonds) activation under light irradiation, so as to achieve high reactivity for oxidation of ethylbenzene with a conversion efficiency of 99.9% in 8 h, which is much higher than imine-based COF. The corresponding details can be found in the revised manuscript and revised supplementary information, as also shown below:

Page 12 in the revised manuscript: “Additionally, we also evaluated the photocatalytic antibacterial efficacy of TTT-COF since H₂O₂ has been widely used for bacterial disinfection. The data suggested that after 30 minutes of light exposure, the antibacterial rate impressively exceeded 99%, which may offer a safe and potent strategy against antibiotic-resistant pathogens (Supplementary Figs. 27-29).”

Page 19 in the revised manuscript: “**Photocatalytic performance toward aerobic oxidation of C(sp³)-H bonds.** The exceptional chemical stability, efficient electron transfer, and prolonged charge carrier lifetimes of thiazole-based COFs position them as promising candidates for photocatalytic aerobic oxidation. We evaluated their performance using ethylbenzene oxidation as a model reaction, revealing striking differences between the fully cyclized TTT-COF and its TTI-COF counterpart. While TTT-COF achieved an excellent acetophenone yield (99%), TTI-COF showed no catalytic activity under identical conditions. Notably, partially cyclized TTT-COF (0.25eq) demonstrated intermediate performance (30% conversion), underscoring the critical role of complete thiazole ring formation in photocatalytic efficiency (Fig. 6a). Control experiments systematically confirmed the essential components of this photocatalytic system: the thiazole-incorporated COF structure, molecular oxygen as oxidant, and appropriate light irradiation all proved indispensable for reaction progression (Supplementary Table 8). These findings highlight how precise structural control over heteroaromatic connectivity in COFs can dramatically influence their photocatalytic performance in aerobic oxidation reactions.

To assess the generality of this photocatalytic system, we examined the oxidation of various ethylbenzene derivatives under standardized conditions (Fig. 6a). TTT-COF maintained exceptional catalytic efficiency for substrates bearing electron-donating halogen substituents (p-F, p-Cl, p-Br), while the electron-withdrawing p-OCH₃ variant showed slightly reduced activity (86% yield). Notably, the framework demonstrated remarkable versatility in converting p-diethylbenzene to the industrially valuable p-diacetylbenzene with 98% yield—a challenging transformation for conventional catalysts. The structural robustness of TTT-COF was evidenced through five consecutive catalytic cycles without

significant activity loss (Fig. 6b), with post-reaction PXRD and FTIR analyses confirming complete retention of crystallinity and chemical functionality (Supplementary Figs. 53, 54). This combination of broad substrate scope and exceptional stability positions thiazole-based COFs as practical candidates for sustainable oxidation catalysis.

In order to explore the catalytic mechanism, we first identified the active species in the reaction process. A series of quenching experiments were performed by adding quenching agent to the original test solution of ethylbenzene oxidation (Fig. 6c). When a hole quencher (KI) or electron quencher (AgNO_3) was added, only a trace of the products could be detected after 8 h light irradiation, suggesting that photogenerated electrons and hole radicals are engaged in the reaction. When a $^1\text{O}_2$ radical quencher (L-Histidine, L-HIS) and $\bullet\text{O}_2^-$ radical quenching agent (p-benzoquinone, BQ) were introduced, the catalytic efficiency was significantly weakened, stating O_2 is reduced to $\bullet\text{O}_2^-$ and $^1\text{O}_2$, and involved in the oxidation reaction. To verify the generation of the above-mentioned ROS, EPR spectroscopy in MeCN of TTT-COF was performed. As shown in Fig. 6d and Supplementary Fig. 55, $\bullet\text{O}_2^-$ and $^1\text{O}_2$ signals appeared in the system after 5 min of illumination, indicating that TTT-COF could rapidly activate oxygen molecules to ROS under light irradiation. However, no $^1\text{O}_2$ was detected in TTI-COF, which may be the main reason why TTI-COF cannot catalyze ethylbenzene oxidation. According to the obtained results, the reaction mechanism for the photocatalytic aerobic oxidation of $\text{C}(\text{sp}^3)\text{-H}$ bonds is proposed in Fig. 6e. The presence of EnT in TTT-COF is attributed to the formation of homologous heteropolyaromatic structure with thiazole rings, which augment the overall conjugation of the framework and facilitates the formation of $^1\text{O}_2$, thereby facilitating efficient photocatalytic performance toward aerobic oxidation of $\text{C}(\text{sp}^3)\text{-H}$ bonds.”

Fig. 6 The photocatalytic performance of TTT-COF for aerobic oxidation of C(sp³)-H bonds. **a** TTT-COF catalyzed the aerobic oxidation of ethylbenzene derivatives. **b** Kinetic profile for oxidation of ethylbenzene. Reaction conditions: photocatalyst (10 mg), ethylbenzene (0.1mmol), xenon lamp (420-1100 nm, 15 W), CH₃CN (3 mL), O₂ (1 atm), 8 h. Determined by GC-FID using chlorobenzene as the internal standard, conversion of ethylbenzene, selectivity of acetophenone. **c** Cycle experiment of aerobic oxidation of ethylbenzene along with TTT-COF as photocatalyst. **d** Control experiments for photocatalytic ethylbenzene oxidation by TTT-COF under normal conditions or with different scavengers. **e** EPR signals of the reaction solution under dark and visible light illumination in the presence of TEMP as the spin-trapping reagent. **f** Proposed mechanism for aerobic oxidation of C(sp³)-

H bonds. Experiments were repeated independently (a-e) three times with similar results. In e, a.u. indicates the arbitrary units. Source data are provided as a Source Data file.

Supplementary Figure 53. PXRD patterns of TTT-COF before and after five photocatalytic oxidation cycles of ethylbenzene (COFs were regenerated by washing with acetone and MeOH), a.u. indicates the arbitrary units.

Supplementary Figure 54. FTIR spectra of TTT-COF before and after five photocatalytic oxidation

cycles of ethylbenzene (COFs were regenerated by washing with acetone and MeOH), a.u. indicates the arbitrary units.

Supplementary Figure 55. a EPR spectra of DMPO·OH generated by TTT-COF and TTI-COF in the dark and under visible light irradiation. (300 W Xenon lamp, 5 min irradiation; O₂ saturated; acetonitrile/ethylbenzene mixture (9/1 v/v, 500 μL)), a.u. indicates the arbitrary units.

(4) *As a material for personal use in bacterial disinfection, the general toxicity of TTT-COF to cells or human should be evaluated.*

Response to comment:

Thanks for your insightful and helpful comments. We agree that as a material for personal use in bacterial disinfection, the general toxicity of TTT-COF to cells or humans should be evaluated. However, considering the experimental conditions and the suggestions of the editor and other reviewers, we have decided not to do further research on materials for personal use in bacterial disinfection. Meanwhile, we evaluated the photocatalytic performance toward aerobic oxidation of C(sp³)-H bonds of COFs in order to enhance the quality and novelty of this work. We have incorporated the new data on aerobic oxidation of C(sp³)-H bonds into the revised manuscript.

Reviewer #4 (Remarks to the Author):

“The manuscript by Cheng and co-workers reports the synthesis of two homologous COFs for photocatalytic H₂O₂ generation and bacterial disinfection. The post-synthetic modification method was used to synthesize thiazole-based homologous heteropolyaromatic TTT-COF from its parent imine-linked TTI-COF via post-cyclization reaction. The resultant TTT-COF exhibited enhanced chemical stability, continuous π -conjugation, excellent electron transfer, and enhanced D–A structure and charge separation, compared to the TTI-COF. The H₂O₂ production rate of TTT-COF was 3.71 mmol g⁻¹ h⁻¹ in pure water under air and further improved to 29.9 mmol g⁻¹ h⁻¹ by utilizing a sacrificial electron donor mixture (9:1 v/v water/benzyl alcohol (BA)) and continuously purging with O₂. The photocatalytic performance of TTT-COF is much superior to TTI-COF, which showed a production rate of only 0.48 mmol g⁻¹ h⁻¹ in pure water and air and 1.4 mmol g⁻¹ h⁻¹ in the presence of sacrificial reagent. Additionally, TTT-COF can be integrated into the polypropylene fiber and personal protection mask, which displayed >99.99% bactericidal efficiency against airborne drug-resistant bacteria within 30 minutes. Though the authors carried out lots of experiments and characterizations, the core novelty of the manuscript is not enough, and the present analysis is insufficient to verify some of their claims. Thus, I cannot recommend its publication.”

Response to the general comment:

We are grateful for the constructive feedback and helpful suggestions on our manuscript. Your comments and suggestions have been instrumental in enhancing the quality of our work. In response to your insights, we have thoroughly revised the manuscript to provide a clearer exposition of the core novelty of the manuscript and the new findings in photocatalytic applications and mechanisms.

First of all, the novelty of this work is not designing a new type of COF. The main purpose of our work is to investigate the effects of the post-cyclization reaction via sublimated sulfur vapor strategy on the optical/optoelectronic properties and corresponding photocatalytic performances of these homologous heteropolyaromatic COFs. Our systematic studies demonstrate that the sublimated sulfur vapor-introduced elemental S heteroatom enables modified COF materials with remarkable chemical stability, continuous π -conjugation, excellent electron and energy transfer, and enhanced D-A structure

and charge separation, thus boosting the intrinsic photocatalytic activities and stabilities. Consequently, TTT-COF achieves a superior photosynthetic H₂O₂ production rate of 29.9 mmol g⁻¹ h⁻¹ with more than 200 hours of long-term stability when employing 10 % benzyl alcohol (V/V) as a sacrificial agent in an O₂-saturated atmosphere, which is a 20-fold improvement over the imine-linked COF and exceeds many of the state-of-the-art photocatalysts. Notably, the TTT-COF photocatalyst can undergo rapid intermolecular migration of photogenerated charge, so as to achieve high reactivity for the oxidation of ethylbenzene derivatives. We wish that these interesting new findings on the optical/optoelectronic properties and photocatalytic performances of these thiazole-based homologous heteropolyaromatic COFs (TTT-COF) via solvent-free and eco-friendly post-cyclization reaction will clarify the novelty of this manuscript.

Furthermore, in the revised manuscript, in order to enhance the strength and significance of the article, we have performed the isotopic labelling experiments in order to clearly investigate the two half reactions mechanism; meanwhile, we have added and evaluated the photocatalytic performances of COFs toward aerobic oxidation of C(sp³)-H bonds. To strengthen the mechanistic analysis reported in the work, we further performed more comprehensive experiments on active intermediates capture and electron paramagnetic resonance. With the above significant revision, we hope we have improved the novelty of this manuscript. If the reviewer has any additional comments, please let us know, we will further enhance this work to meet the requirements.

Secondly, we conducted the following experiments to complement and refine the study of the photocatalytic mechanism:

1. Electron paramagnetic resonance (EPR) measurements
2. Isotopic labelling experiments
3. Active intermediates capture

All queries and concerns have been addressed in the revised manuscript and the accompanying supplementary material. Your guidance has been invaluable in refining our study, and we are confident that these revisions have substantially improved the paper. Once again, we extend our thanks for your insightful contributions.

(1) *The COF synthesis is simple and has been reported previously, which largely limited the novelty of this work. TTI-COF produced by imine polycondensation and TTT-COF generated by post modification are quite straightforward. TTI-COF has been reported by many research groups (Processes 2023, 11(10), 2859, ACS Materials Lett. 2024, 6, 11, 5016–5022, Mater. Chem. Front., 2017,1, 1354-1361, Chem. Eur. J. 2020, 26, 5583).*

Thank you for your good comments and helpful suggestions. We acknowledge that TTI-COF has been reported by other research groups. However, the main purpose of our paper is to investigate the effect of the post-cyclization reactions on the optical properties and photocatalytic performances of homologous heteropolyaromatic COFs, which have not been studied by previous research. Therefore, we chose a COF with a relatively simple structure for the post-cyclization reaction in order to better study the changes in optical properties caused by structural changes. The results show that the performance was significantly improved after cyclization. Meanwhile, we have also investigated the reasons for the performance improvement is that continuous π -conjugation of TTT-COF guarantees excellent single electron transfer (SET) and energy transfer (EnT) for photocatalysis, and the thiazole-based frameworks can significantly enhance the intramolecular polarity, D-A structure, and charge separation, thus boosting the intrinsic photocatalytic activities.

In addition, in order to further prove that the cyclization reaction has a promoting effect on optical properties, we also evaluated the performance of COFs on the photocatalytic aerobic oxidation. The result shows that the constructed photocatalyst, TTT-COF, can undergo rapid intermolecular migration of photogenerated charges and substrate (C(sp³)-H bonds) activation under light irradiation, so as to achieve high reactivity for oxidation of ethylbenzene with a conversion efficiency of 99.9% in 8 h, which is much higher than imine-based COF.

Furthermore, in the revised manuscript, in order to enhance the strength and significance of the article, we have performed the isotopic labelling experiments in order to clearly investigate the two half-reaction mechanisms. To strengthen the mechanistic analysis reported in the work, we further performed more comprehensive experiments on active intermediates capture and electron paramagnetic resonance. With the above significant revision, we hope we have improved the novelty

of this manuscript. If the reviewer has any additional comments, please let us know. We will further enhance this work to meet the requirements.

(2) *The proposed oxidative cyclization synthetic strategy for TTT-COF is reported by other groups (J. Am. Chem. Soc. 2018, 140, 29, 9099–9103, Nat Commun 2018, 9, 2600), which should be clearly cited and demonstrated in the introduction. However, the manuscript provided a vague statement and citation, which is misleading and harmful to the research community.*

Response to comment:

We are grateful for the constructive feedback on our manuscript. Your comments and suggestions have been instrumental in enhancing the quality of our work. This is indeed an oversight on our part for not explicitly citing previous research in the introduction. We agree with the reviewer that the sulfur vapor strategy used in this work was inspired by the *J. Am. Chem. Soc. 2018, 140, 29, 9099–9103*, and *Nat. Commun.*, **2018**, 9, 2600.

We appreciate your thoughtful input, which has significantly informed our work. The corresponding details can be found in the revised manuscript, as also shown below:

Page 4 in the revised manuscript: “Though recent research has proposed synthesizing the thiazole-based homologous polyaromatic COFs via sulfur vapor strategy^{77,78}, the effects and mechanisms of the post-cyclization reaction via sublimated sulfur atoms on the optical/optoelectronic properties and corresponding photocatalytic applications of these homologous polyaromatic COFs have not been disclosed.

Here, we proposed the design of thiazole-based homologous heteropolyaromatic COFs (TTT-COF) via solvent-free and eco-friendly post-cyclization reaction for superior photocatalytic H₂O₂ production and aerobic oxidation of C(sp³)-H bonds.”

References:

- 77 Waller, P. J., AlFaraj, Y. S., Diercks, C. S., Jarenwattananon, N. N. & Yaghi, O. M. Conversion of Imine to Oxazole and Thiazole Linkages in Covalent Organic Frameworks. *J. Am. Chem. Soc.* **140**, 9099-9103 (2018).
- 78 Haase, F. *et al.* Topochemical conversion of an imine- into a thiazole-linked covalent organic

framework enabling real structure analysis. *Nat. Commun.* **9**, 2600 (2018).

Additionally, the general synthetic method and principle are the same as earlier work, and the specific experimental difference is that we transferred the homogeneous mixture to a Pyrex tube instead of a quartz boat, and then the mixture was evacuated under vacuum, and purged with argon gas in a cycle repeated three times. With this treatment, S vapor does not leak and interacts better with the TTT-COF, greatly improving the imine-to-thiazole linkages conversion ratios. Meanwhile, in order to determine the optimal S content during the post-cyclization reaction, TTI-COF samples were subjected to treatments with varying sulfur dosages.

Moreover, the main purpose of our paper is to investigate the effect of the post-cyclization reaction on the optical properties and photocatalytic performance of COFs, which is different from the previous work. Through the characterization of optoelectronic properties, the performance test on photocatalytic H₂O₂ production, the test of photocatalytic aerobic oxidation and the calculation of DFT, the following conclusions are obtained: this TTT-COF demonstrates three unique structural advantages (Fig. 1): 1) introducing the S heteroatom to form heteropolyaromatic COF enables remarkable chemical stability; 2) the unique structure with continuous π -conjugation of TTT-COF guarantees excellent single electron transfer (SET) and energy transfer (EnT) for photocatalysis; 3) the thiazole-based frameworks can significantly enhance the intramolecular polarity, D-A structure, and charge separation, thus boosting the intrinsic photocatalytic activities.

(3) *The method for synthesis of TTT-COF is not clear. Whether the oxidation reaction was conducted under the oxygen atmosphere or not is unknown. Why was the reaction temperature maintained at 155 °C for 3 h before increasing to 350 °C? What is the yield of TTT-COF?*

Response to comment:

We are grateful for your insightful comments, and we are sincerely sorry for the unclear statements in our earlier manuscript. This post-cyclization reaction was conducted in an argon atmosphere. During the reaction, sulfur serves as an oxidant (being reduced to H₂S) and as a nucleophile, attaching first to the imine carbon and afterwards to the phenyl ring on the nitrogen side of the imine. The reason why

the reaction temperature is maintained at 155 °C for 3 h before increasing to 350 °C is that at this temperature, the sulfur melts and exhibits minimal viscosity, which then can be homogeneously mixed with TTI-COFs. The reaction can be fully carried out only after thoroughly mixing and evenly reaching the reaction temperature of 350 °C. The yield ratio of TTT-COF is 90%, because some losses of products inevitably occur during the reaction, such as in the process of grinding, transferring vessels, and so on. We also systematically studied the imine-to-thiazole linkage conversion ratio during the reaction. We found that the imine bond can be fully converted under optimized reaction conditions. Please see the detailed responses to Comment 1 of Reviewer #3. The corresponding details can be found in the revised manuscript, as also shown below:

The synthesis method in our work is as follows: The activated imine-linked TTI-COF (100 mg) was intimately mixed with molecular sulfur (1.5 g) using a mortar and pestle. After grinding for 5 minutes, the homogeneous mixture was transferred to a Pyrex tube, evacuated under vacuum, and purged with argon gas in a cycle repeated three times. The sealed tube was then placed in a furnace, and the temperature was ramped up to 155 °C at a rate of 2 °C per minute and held at this temperature for 3 hours. The temperature was further increased to 350 °C, maintaining the same heating rate, and kept at this elevated temperature for an additional 3 hours. Upon natural cooling, the brown-colored product was extracted using a Soxhlet apparatus with toluene and tetrahydrofuran for 12 hours each. The final product, COF powder, was washed with methanol and dried under vacuum at room temperature to obtain the TTT-COF (90 mg, 90%). The TTT-xeq materials were prepared using the same procedure, referring to different sulfur dosages mixed with TTI-COF at mass ratios of 0.05, 0.25, 0.5, 1, 5, 10, and 15 using a mortar and pestle.”

(4) Before and after modification, the FTIR and ¹³C ssNMR spectra seem very similar, which may be due to the weak signals. This is difficult for claiming the formation of thiazole-linked TTT-COF. The corresponding characterizations of model compounds should be provided for validation.

Response to comment:

We sincerely appreciate your insightful and constructive comments aimed at enhancing the quality of our manuscript. It is a good idea to provide model compounds to demonstrate the conversion of COF. We have synthesized the (E)-2-((phenylimino)methyl) phenol molecule and also post-cyclized it

in a furnace. However, this reaction did not proceed successfully, and it is possible that the small molecule had broken down at 350 °C and could not be post-cyclized.

The chemical formula of (E)-2-((phenylimino)methyl)

The image of (E)-2-((phenylimino)methyl) phenol molecule after post-cyclization in a furnace.

However, to reveal that the formation of thiazole-linked TTT-COF has occurred in this work, we investigate the conversion ratios of the imine linkages to the thiazole linkages by changing the sulfur dosages, the TTT-xeq materials were prepared using the same procedure as TTI-COF, referring to different sulfur dosages mixed with TTI-COF at mass ratios of 0.05, 0.25, 0.5, 1, 5, 10, and 15 using a mortar and pestle. Then, the obtained different samples were characterized by elemental analysis (EA), Fourier-transform infrared spectroscopy (FTIR), and powder X-ray diffraction (PXRD). As can be seen from the FTIR, the imine vibration located at 1627 cm^{-1} completely disappears when the sulfur dosage is 0.5 times the mass of the TTI-COF. Meanwhile, EA tests show the presence of sulfur with an elemental composition close to the composition that would be expected from the thiazole unit. Thus,

we can conclude that the conversion from imine bonds to the thiazole molecule has been nearly completed when the sulfur dosage is 0.5 times the mass of the TTI-COF. However, when the sulfur dosage increases, PXRD data show that the crystallinity of TTT-COF also increases significantly. Therefore, when considering both the crystallization and the complete conversion, we choose the 15 times sulfur dosage to the mass of the TTI-COF, which can guarantee both the full conversion of imine bonds and high crystallinity of TTT-COF.

In summary, through the exploration of synthesis conditions, we found that TTI-COF could be completely transformed into highly crystallized TTT-COF at 15 times the sulfur dosage to the mass of the TTI-COF. In contrast, its imine bonds can only be partially transformed when the sulfur dosage is lower than 0.5 times the mass of the TTI-COF. The corresponding details can be found in the revised manuscript and revised supplementary information, as also shown below:

Supplementary Table 5. Elemental analysis of different TTT-COF samples obtained from treating TTI-COF with varying sulfur dosages.

	C	N	H	S
Expected	68.95	16.08	2.70	12.27
TTT-0.05eq	76.51	17.06	3.58	2.85
TTT-0.25eq	71.76	14.32	3.13	10.79
TTT-0.5eq	68.54	15.87	2.87	12.72
TTT-15eq	68.17	15.46	2.73	13.64

Supplementary Figure 11. Fourier-transform infrared spectroscopy (FTIR) spectra of TTT-COF treated with varying sulfur dosage, a.u. indicates the arbitrary units. FTIR analysis demonstrated a partial to complete conversion from imine bonds to thiazole bonds across sulfur dosages ranging from 0.05 to 15 times the mass of TTI-COF. PXRD characterization confirmed that the optimal sulfur dosage (15 times the mass of TTI-COF) achieved balanced cyclization efficiency and crystallinity preservation.

Supplementary Figure 10. Powder X-ray diffraction (PXRD) patterns of TTT-COF treated with varying sulfur dosage, a.u. indicates the arbitrary units. The crystallinity of TTT-COF exhibited a

progressive enhancement with increasing sulfur dosage, demonstrating sulfur's role in modulating the crystallization behavior of the COF during the synthetic process.

(5) *In Figure 3, TTT-COF exhibited stronger photocurrent signals and smaller charge transfer resistance than TTI-COF. Any reasons for this enhancement?*

Response to comment:

Thanks for your insightful and helpful comments. As depicted in our main text, the unique structure with continuous π -conjugation of TTT-COF guarantees excellent electron transfer for photocatalysis, and the thiazole-based frameworks can significantly enhance the intramolecular polarity, D-A structure, and charge separation, thus boosting the intrinsic photocatalytic activities. Therefore, TTT-COF exhibited stronger photocurrent signals and smaller charge transfer resistance than TTI-COF. The experimental results show that TTT-COF has a smaller band gap, which makes it easier for electrons to be excited, and correspondingly has a stronger photocurrent response. Meanwhile, we also conducted density functional theory (DFT) calculations in order to investigate further the change of the D-A structure. Total charge numbers of different components in TTI-COF and TTT-COF were calculated, 1.39 |e| for TTI-COF and 1.79 |e| for TTT-COF, respectively, to evaluate their intramolecular polarities. Such enhanced intramolecular polarity boosted the electron donor-acceptor effect and facilitated the charge separation in TTT-COF. Moreover, as depicted in the electron localization functions (ELF) diagrams, TTT-COF has an additional electron transport channel, which extends the delocalization of π electrons, resulting in enhanced charge separation. The corresponding details have been added in the revised manuscript and revised supplementary information, as also shown below:

Page 4 in the revised manuscript: “As illustrated in Fig. 1, our research is driven by three primary purposes: 1) the introducing S heteroatom to form heteropolyaromatic COF enables remarkable chemical stability (Fig. 1a); 2) the unique structure with continuous π -conjugation of TTT-COF guarantees excellent **single** electron transfer (SET) and **energy transfer (EnT)** for photocatalysis; 3) the thiazole-based frameworks can significantly enhance the intramolecular polarity, D-A structure, and charge separation (Figs. 1b, c), thus boosting the intrinsic photocatalytic activities.”

Page 9 in the revised manuscript: “Following the confirmation of the synthesis of TTI-COF and TTT-COF through comprehensive characterization, we proceeded to investigate their light absorption properties and energy band structures. UV–vis diffuse reflectance spectra (DRS) revealed effective light absorption in the visible spectrum for both materials (Fig. 3a). Notably, TTT-COF exhibited a more extensive absorption range from 450 nm to 600 nm, attributable to its enhanced conjugation afforded by the thiazole linkages. This absorption profile corresponded well with the yellowish hue of TTI-COF and the brownish tint of TTT-COF as disclosed in Fig. 1a. The optical band gaps (E_g) were determined to be 2.83 eV for TTI-COF and were slightly reduced to 2.66 eV for TTT-COF, as derived from Tauc plots using the Kubelka–Munk equation from DRS analysis (Fig. 3a). The flat band potentials were ascertained to be -1.00 V for TTI-COF and -1.13 V for TTT-COF using Mott–Schottky plots (Supplementary Fig. 15). By integrating the optical band gaps with the Mott–Schottky data, the band structure configurations were deduced (Fig. 3b). The conduction bands (CB) of TTI-COF and TTT-COF were calculated to be -0.80 V and -0.93 V (vs. normal hydrogen electrode, *NHE*), respectively, based on the flat band potentials. Subsequently, the valence bands (VB) were estimated to be 2.03 V and 1.73 V (vs. *NHE*) using the relationship $E_{CB} = E_{VB} - E_g$. Given that the CB positions of both TTI-COF and TTT-COF are more negative than the redox potentials for the $2e^-$ oxygen reduction reaction (ORR, -0.33 V for indirect ORR, and 0.68 V for direct ORR at pH=0, vs. *NHE*), they are thermodynamically capable of producing H_2O_2 via either a direct or indirect $2e^-$ ORR pathway.”

Page 16 in the revised manuscript: “To investigate the origin of the catalytic activity of TTT-COF, density functional theory (DFT) calculations were conducted.^{80,81} Total charge numbers of different components in TTI-COF and TTT-COF were first calculated, 1.39 |e| for TTI-COF and 1.79 |e| for TTT-COF, respectively, to evaluate their intramolecular polarities. Such enhanced intramolecular polarity boosted the electron donor-acceptor effect and facilitated the charge separation in TTT-COF (Supplementary Fig. 47). Moreover, the electron localization functions (ELF) diagrams and the charge density differences of TTI-COF and TTT-COF were illustrated (Figs. 5e, f and Supplementary Fig. 48). Specifically, introducing the S atom altered the charge state of adjacent C/H moieties and extended the delocalized pathways for π electrons, leading to the superior intrinsic activity of TTT-COF.”

Fig. 3 Optical and optoelectronic properties for photocatalytic COFs. **a** Solid-state UV-Vis diffuse reflectance spectrum and Tauc plot for band gap calculation. **b** Band-structure diagrams of TTI-COF and TTT-COF. **c** Photocurrent-responses of TTI-COF and TTT-COF.

Supplementary Figure 15. Mott-Schottky plots of (a) TTI-COF and (b) TTT-COF. ECB (V vs. NHE) was calculated according to the formula: $E(\text{NHE}) = E(\text{Ag}/\text{AgCl}) + 0.197$.

Supplementary Figure 47. The calculated charge distribution of (a) TTI-COF and (b) TTT-COF structures. (color code: orange, S; gray, C; sky blue, N; and white, H).

Supplementary Figure 48. The calculated Electron Localization Function (ELF) diagrams of (a) TTI-COF and (b) TTT-COF.

80 Kresse, G. & Furthmüller, J. Efficiency of ab-initio total energy calculations for metals and semiconductors using a plane-wave basis set. *Comput. Mater. Sci* **6**, 15-50 (1996).

81 Kresse, G. & Furthmüller, J. Efficient iterative schemes for ab initio total-energy calculations using a plane-wave basis set. *Phys. Rev. B* **54**, 11169-11186 (1996).

(6) TTT-COF displayed pronounced photoluminescence quenching and longer fluorescence lifetime compared with TTI-COF. Why does increasing conjugation of COF lead to these variations?

Response to comment:

Thank you for your attentive review and valuable suggestions. The reason why TTT-COF can lead to pronounced photoluminescence quenching and longer fluorescence lifetime is that conjugate expansion narrows the band gap and promotes the dissociation of excitons into free carriers, and the long-range ordered structure provides a charge transport channel and reduces the probability of radiative recombination (*Chem. Rev.*, **2020**, 120, 8814–8933; *Angew. Chem. Int. Ed.*, **2025**, e202501869; *Angew. Chem. Int. Ed.*, **2024**, 63, e202408802). In addition, the increase of conjugation of COF can enhance spin-orbit coupling, accelerate the process of electron intersystem crossing (ISC), and promote triplet exciton generation (*Adv. Mater.*, **2025**, 2502220). In our paper, the conversion of imine linkage to thiazole linkage forms a more efficient D-A structure, which improves the charge separation efficiency and weakens the photogenerated charge recombination, so that TTT-COF displayed pronounced photoluminescence quenching and longer fluorescence lifetime compared with TTI-COF. Meanwhile, the results of our calculations show that increasing conjugation of COF can promote greater spatial separation characteristics, which partially suppresses electron-hole recombination, thus prolonging electron/hole lifetimes. The corresponding details have been added in the revised manuscript and revised supplementary information, as also shown below:

Page 17 in the revised manuscript: “Moreover, the electron localization functions (ELF) diagrams and the charge density differences of TTI-COF and TTT-COF were illustrated (Figs. 5e, f and Supplementary Fig. 48). Specifically, introducing the S atom altered the charge state of adjacent C/H moieties and extended the delocalized pathways for π electrons, leading to the superior intrinsic activity of TTT-COF. As shown in Supplementary Fig. 49, the orbital distribution of the conduction band minimum (CBM) and valence band maximum (VBM) indicates that TTT-COF exhibits greater spatial separation characteristics, which partially suppresses electron-hole recombination, prolongs electron/hole lifetimes, and facilitates the progress of photocatalytic reactions.”

(7) Though the proposed TTT-COF exhibited improved photocatalytic H₂O₂ production rate compared to the imine-linked TTI-COF, the overall photocatalytic performance of TTT-COF in either pure water or water/BA mixture is quite average in comparison to other reported photocatalytic COFs (*Angew. Chem. Int. Ed.* 2023, 62, e202218868, *J. Am. Chem. Soc.* 2024, 146, 20107–20115, *Angew. Chem. Int. Ed.* 2025, 64, e202416350).

Response to comment:

Thank you for your valuable and constructive comments. The novelty of this work is not designing a new type of COF. The main purpose of our work is to investigate the effects of the post-cyclization reaction via sublimated sulfur vapor strategy on the optical/optoelectronic properties and corresponding photocatalytic performances of these homologous heteropolyaromatic COFs. Considering that homologous polyaromatic COFs exhibit a higher degree of structural similarity and uniformity, which facilitates not only the synthesis and structural characterization but also a more coherent analysis of the structure and function of their underlying photocatalytic mechanisms. Therefore, we chose a COF with a homologous structure for the post-cyclization reaction to better study the changes in optical properties caused by structural changes.

Our systematic studies demonstrate that the sublimated sulfur vapor-introduced elemental S heteroatom enables modified COF materials with remarkable chemical stability, continuous π -conjugation, excellent electron and energy transfer, and enhanced D-A structure and charge separation, thus boosting the intrinsic photocatalytic activities and stabilities. Consequently, TTT-COF achieves a superior photosynthetic H₂O₂ production rate of 29.9 mmol g⁻¹ h⁻¹ with more than 200 hours of long-term stability when employing 10 % benzyl alcohol (V/V) as a sacrificial agent in an O₂-saturated atmosphere, which is a 20-fold improvement over the imine-linked COF and exceeds many of the state-of-the-art photocatalysts. Through careful comparison with the literature, including the ones listed here (Supplementary Table 7, also can be seen in the response to comment 2 of Reviewer #3), we found that this TTT-COF exhibited the highest H₂O₂ photo-production rate when using 10% benzyl alcohol as sacrificial reagents.

Furthermore, in the revised manuscript, in order to enhance the strength and significance of the article, we have performed the isotopic labelling experiments in order to clearly investigate the two

half reactions mechanism; meanwhile, we have added and evaluated the photocatalytic performances of COFs toward aerobic oxidation of C(sp³)-H bonds. To strengthen the mechanistic analysis reported in the work, we further performed more comprehensive experiments on active intermediates capture and electron paramagnetic resonance. We wish that these interesting new findings on the optical/optoelectronic properties and photocatalytic performances of these thiazole-based homologous heteropolyaromatic COFs (TTT-COF) via solvent-free and eco-friendly post-cyclization reaction will clarify the novelty of this manuscript.

(8) The authors confirmed the superoxide anion radical as possible intermediate, but the photoactivation of oxygen usually accompanied with the production of singlet oxygen. Whether this active oxygen species involved in the photocatalytic reaction is not studied.

Response to comment:

Thank you for your constructive remarks on our manuscript. We wholeheartedly agree that studying singlet oxygen is essential for mechanistic studies. In response to your feedback, we have conducted electron paramagnetic resonance (EPR) measurements to study singlet oxygen. EPR measurements were performed in H₂O using 2,2,6,6-Tetramethyl-4-piperidone (TEMP) as a free-radical spin-trap agent. The results show that the EPR spectra of TTT-COF reaction systems exhibited a weak signal indicative of singlet oxygen (¹O₂) under visible light irradiation, confirming that there may be a small amount of ¹O₂ involved in the photocatalytic process and that the energy transfer (EnT) process may have occurred. In contrast, no signal was detected in TTI-COF, confirming that imine linkages with discontinuous π-conjugation impede the EnT process. The corresponding details have been added in the revised manuscript and revised supplementary information, as also shown below:

Page 15 in the revised manuscript: “Meanwhile, using 2,2,6,6-tetramethyl-4-piperidone (TEMP) as a singlet oxygen (¹O₂) spin-trapping agent, the EPR spectra of TTT-COF reaction systems exhibited a weak signal indicative of ¹O₂ under visible light irradiation. In contrast, no signal was detected in TTI-COF, confirming that imine linkages with discontinuous π-conjugation impedes EnT process (Supplementary Fig. 43).”

Supplementary Figure 43. Electron paramagnetic resonance (EPR) signals of the reaction solution under dark and light irradiation in the presence of 2,2,6,6-tetramethyl-4-piperidone (TEMP) as the spin-trapping reagent. (300 W Xenon lamp, 5 min irradiation; O₂ saturated; H₂O), a.u. indicates the arbitrary units.

(9) *The mechanism investigation is not completed. The photocatalytic H₂O₂ production usually involves two half-reactions of oxygen reduction and water oxidation. The photocatalytic production of H₂O₂ can also go through the •OH involved two-step WOR process (Angew. Chem. Int. Ed. 2022, 61, e202200413) and the direct oxidation of water to H₂O₂, but the author did not investigate these pathways, so the proposed reaction mechanism is doubtful.*

Response to comment:

Thanks for your important and helpful comments on improving the quality of our manuscript. We agree with your points that the WOR process should be studied to complete the mechanism investigation. To fully elucidate the photocatalytic mechanism, we have conducted a series of additional experiments, including active intermediates capture, electron paramagnetic resonance (EPR), and isotope labeling measurements. The result implies that a four-electron water oxidation reaction (WOR) has occurred in TTT-COF ($2\text{H}_2\text{O} + 4\text{h}^+ \rightarrow \text{O}_2 + 4\text{H}^+$), and the •OH is not involved in

the WOR process. The corresponding details can be found in the revised manuscript and revised supplementary information, as also shown below:

Page 15 in the revised manuscript: “To elucidate the photocatalytic mechanism underlying H₂O₂ production in this study, we conducted a series of experiments, including active intermediates capture, electron paramagnetic resonance (EPR), in-situ diffuse reflectance infrared Fourier transform spectroscopy (DRIFTS), and isotope labeling measurements. The introduction of benzoquinone (BQ, a •O₂⁻ scavenger) significantly suppressed H₂O₂ production, while the addition of t-butyl alcohol (TBA), a hydroxyl radical (•OH) scavenger, had no effect on H₂O₂ generation (Fig. 5a). These findings indicate that H₂O₂ generation in this reaction system is predominantly dependent on electron transfer and necessitates the involvement of •O₂⁻ rather than •OH. Consequently, the photoproduction of H₂O₂ is primarily associated with a two-step single-electron oxygen reduction reaction (ORR) process: O₂ + e⁻ → •O₂⁻, •O₂⁻ + 2H⁺ + e⁻ → H₂O₂. Additionally, when the holes were trapped in the presence of CH₃OH and O₂, the H₂O₂ production for TTT-COF presented an upward trend. This phenomenon indicates that holes generated from TTT-COF may not be directly involved in the photocatalytic production of H₂O₂. Meanwhile, the H₂O₂ concentration was almost undetectable for TTT-COF when the electron-trapping agent (AgNO₃) was added in the presence of N₂. This result implies that a four-electron water oxidation reaction (WOR) may have occurred in TTT-COF (2H₂O + 4h⁺ → O₂ + 4H⁺).

To confirm the presence of •O₂⁻ in the ORR pathway, EPR measurements were performed in methanol/H₂O using 5,5-dimethyl-1-pyrroline N-oxide (DMPO) as a free-radical spin-trap agent. As depicted in Fig. 5b, upon visible light irradiation, the EPR spectra of both TTI-COF and TTT-COF reaction systems exhibited characteristic signals indicative of •O₂⁻. In contrast, no signals were detected in the absence of light, confirming the formation of •O₂⁻ within the photocatalytic system. Meanwhile, using 2,2,6,6-tetramethyl-4-piperidone (TEMP) as a singlet oxygen (¹O₂) spin-trapping agent, the EPR spectra of TTT-COF reaction systems exhibited a weak signal indicative of ¹O₂ under visible light irradiation. In contrast, no signal was detected in TTI-COF, confirming that imine linkages with discontinuous π-conjugation impedes the EnT process (Supplementary Fig. 43). However, the presence of •OH was not detected in the EPR measurements, which rules out the possibility of the 1e⁻ WOR process during the H₂O₂ photosynthesis in TTT-COF (Supplementary Fig. 44).

Fig. 5 Reaction pathways and mechanisms of H₂O₂ photosynthesis. a Photocatalytic H₂O₂ production for TTT-COF in neat water and water with tert-butyl alcohol (TBA), methanol (MeOH), AgNO₃, and benzoquinone (BQ) (50 mL 10 mM aqueous solution, 25 mg COF), all with 1 h illumination.

Supplementary Figure 43. Electron paramagnetic resonance (EPR) signals of the reaction solution under dark and light irradiation in the presence of 2,2,6,6-tetramethyl-4-piperidone (TEMP) as the spin-trapping reagent. (300 W Xenon lamp, 5 min irradiation; O₂ saturated; H₂O), a.u. indicates the arbitrary units.

Supplementary Figure 44. Electron paramagnetic resonance (EPR) signals of the reaction solution under dark and light irradiation in the presence of 5,5-dimethyl-1-pyrroline N-oxide (DMPO) as the spin-trapping reagent. (300 W Xenon lamp, 5 min irradiation; O₂ saturated; H₂O), a.u. indicates the arbitrary units.

Moreover, we conducted experiments using isotope labeling to further demonstrate the 4e⁻ WOR process. Indeed, before we carry out this experiment, to ensure our reliability of our experimental conditions, we have already reviewed abundant existing literature (*Nat. Commun.*,**2023**, 14, 5238; *J. Am. Chem. Soc.*, 2024, 146, 29943-2995; *Angew. Chem. Int. Ed.*,**2024**, 63, e202410179; *Nat. Catal.*, **2024**, 7, 195-206). We finally settled on two protocols, using ¹⁸O₂ and H₂¹⁸O in two separate batches of reaction. The specific protocol is as follows:

1. Photocatalysts (5 mg) and pure water (H₂¹⁶O, 2 mL) were added to a 10 mL glass reactor. The reaction mixture was deoxygenated 3 times and backfilled with highly pure N₂ (99.999%). 5 mL ¹⁸O₂ was injected into the reactor via syringe, and the reaction was performed for 4 h under the irradiation of a 300 W xenon lamp at ambient temperature. In a separate 5 mL reactor, 100 mg of MnO₂ was introduced, and Ar was purged to eliminate O₂. The light-induced H₂O₂ solution was then added to the reactor containing MnO₂ to generate O₂, which was eventually quantified using a mass spectrometer.

2. Catalysts (5 mg) and H₂¹⁸O (98%, 2 mL) were put into a hermetic device mainly composed of a quartz tube and sealing components (the air was pumped away with a vacuum pump). After O₂ was bubbled into the suspension in the dark for 30 min, the suspension was stirred in the dark for 30 min to reach the absorption-desorption equilibrium. After 36 h of irradiation, the gas products in the headspace of the reaction vessel were analyzed by GC-MS. Meanwhile, the formed H₂O₂ was decomposed by MnO₂ under an Ar atmosphere in another reactor. The O₂ generated by the decomposition of photogenerated H₂O₂ was analyzed by GC-MS.

Eventually, the experimental results showed that in the sealed photocatalytic reaction system with ¹⁸O-labeled oxygen molecules, a higher ratio of ¹⁸O₂ and ¹⁶O₂ for TTT-COF was observed, indicating that H₂¹⁸O₂ was the dominant product that came from the reduction of ¹⁸O₂. Meanwhile, when the photocatalytic reaction was conducted in pure H₂¹⁸O, ¹⁸O₂ was detected directly from the headspace after photoirradiation, confirming the 4e⁻ WOR process in TTT-COF. The corresponding details can be found in the revised manuscript and revised supplementary information, as also shown below:

Page 16 in the revised manuscript: “In addition, isotope labeling experiments were performed using ¹⁸O₂ and H₂¹⁸O, in two separate batches of reaction, as shown in Fig. 5c and Supplementary Fig. 45. In the sealed photocatalytic reaction system with ¹⁸O-labeled oxygen molecules, the content of ¹⁸O₂ and ¹⁶O₂ was detected via decomposition of formed H₂O₂, a higher ratio of ¹⁸O₂ and ¹⁶O₂ for TTT-COF was observed, indicating that H₂¹⁸O₂ was the dominant product that came from the reduction of ¹⁸O₂ (Fig. 5c). Furthermore, when the photocatalytic reaction was conducted in pure H₂¹⁸O, ¹⁸O₂ was detected directly from the headspace after photoirradiation, confirming the 4e⁻ WOR process in TTT-COF. Thereafter, we extracted the aqueous phase, and then injected it into a vial containing MnO₂. ¹⁸O₂ was detected directly from the GC-MS spectrometer, which revealed that the generated O₂ from WOR could subsequently participate in the ORR to form H₂O₂ (Supplementary Fig. 45).”

Fig. 5 Reaction pathways and mechanisms of H_2O_2 photosynthesis. **c** $^{18}\text{O}_2$ isotope labeling experiment for TTT-COF to explore the source of H_2O_2 .

Supplementary Figure 45. H_2^{18}O isotope experiment on TTT-COF to explore water oxidation reaction (WOR) half reaction, a.u. indicates the arbitrary units.

(10) The catalytic sites were not investigated. Which molecular unit was used for oxygen reduction reaction? Which structure was used for oxidation half-reaction? The author should provide more explanations on Figure 5i and 5j.

Response to comment:

We sincerely appreciate your valuable feedback, which has allowed us to further improve the quality of this paper. The oxygen reduction reaction occurs at the electron acceptor section (the thiazole unit), and water oxidation or benzyl alcohol oxidation occurs at the electron donor section. The corresponding details can be found in the revised manuscript, as also shown below:

Page 17 in the revised manuscript: “As shown in Fig. 5i, the oxygen reduction reaction occurs at the electron acceptor section (the thiazole unit), and water oxidation or benzyl alcohol oxidation occurs at the electron donor section. As depicted in Fig. 5j, which illustrates the free energy profile for the specific process of photocatalytic reduction of O₂ to H₂O₂, both TTI-COF and TTT-COF exhibit good photocatalytic properties; notably, the TTT-COF shows a particularly stronger intrinsic activity (-0.17 vs -0.20 eV). Therefore, TTT-COF demonstrates better photocatalytic activities, which is consistent with the above experimental observations.”

Fig. 5 Reaction pathways and mechanisms of H₂O₂ photosynthesis. i Schematic mechanism on photocatalytic H₂O₂ production of TTT-COF. **j** The calculated energy profiles for the reduction of O₂ into H₂O₂ of TTI-COF and TTT-COF.

(11) In Figure 5g and Supplementary Figure 35, the author suggested that the doping of S atoms can introduce active electrons near the Fermi level in TTT-COF. It is difficult to observe the difference from their PDOS. More detailed explanations should be provided.

Response to comment:

We sincerely appreciate the reviewers' insightful comments. To enable systematic comparison, the pDOS of both TTI-COF and TTT-COF have been consolidated into a unified plot to evaluate the effects induced by S atom incorporation. As demonstrated in Fig. 5g and Supplementary Fig. 50, the introduction of S atoms indeed generates new electronic states proximal to the Fermi level. However, considering that S does not directly participate in the H₂O₂ production pathway but rather indirectly modulates the electronic distribution of adjacent C/N atoms to enhance intramolecular polarity, we have accordingly refined the relevant descriptions as follows:

Page 17 in the revised manuscript: “Furthermore, compared to TTI-COF, the doping of S atoms introduces additional electronic states near the Fermi level (Fig. 5g and Supplementary Fig. 50), which induce a redistribution of electronic configurations in adjacent C/N atoms, thereby enhancing intramolecular polarity.”

Supplementary Figure 50. The computed pDOS of TTT-COF and TTI-COF.

(12) *The use of COFs for bacterial disinfection via photocatalytic generation of reactive oxygen species is not new (Angew. Chem. Int. Ed. 2022, 61, e202200413, Angew. Chem. Int. Ed. 2024, 63, e202318562).*

Response to comment:

Thank you for your invaluable feedback and thoughtful comments on our manuscript. We agree with the reviewer that the use of COFs for bacterial disinfection via photocatalytic generation of reactive oxygen species is not new. Before conducting this experiment, we also reviewed abundant existing literature, so we didn't aim to innovate in this experiment. We conducted bacterial disinfection experiments to demonstrate the antimicrobial potential of photocatalytic generation of reactive oxygen species. Meanwhile, considering the suggestions from the editor and other reviewers, we have decided to remove the bacterial disinfection applications to the supporting information in order to make the overall paper focus on the photocatalysis. Meanwhile, we have added the photocatalytic performance toward aerobic oxidation of C(sp³)-H bonds of COFs in order to enhance the quality and novelty of this work. The results show that the constructed photocatalyst TTT-COF can undergo rapid intermolecular migration of photogenerated charges and substrate (C(sp³)-H bonds) activation under light irradiation, so as to achieve high reactivity for oxidation of ethylbenzene with a conversion efficiency of 99.9% in 8 h, which is much higher than imine-based COF. The corresponding details can be found in the revised manuscript and revised supplementary information, as also shown below:

Page 12 in the revised manuscript: “Additionally, we also evaluated the photocatalytic antibacterial efficacy of TTT-COF since H₂O₂ has been widely used for bacterial disinfection. The data suggested that after 30 minutes of light exposure, the antibacterial rate impressively exceeded 99%, which may offer a safe and potent strategy against antibiotic-resistant pathogens (Supplementary Figs. 27-29).”

Page 19 in the revised manuscript: “**Photocatalytic performance toward aerobic oxidation of C(sp³)-H bonds.** The exceptional chemical stability, efficient electron transfer, and prolonged charge carrier lifetimes of thiazole-based COFs position them as promising candidates for photocatalytic aerobic oxidation. We evaluated their performance using ethylbenzene oxidation as a model reaction, revealing striking differences between the fully cyclized TTT-COF and its TTI-COF counterpart. While TTT-COF achieved an excellent acetophenone yield (99%), TTI-COF showed no catalytic

activity under identical conditions. Notably, partially cyclized TTT-COF (0.25eq) demonstrated intermediate performance (30% conversion), underscoring the critical role of complete thiazole ring formation in photocatalytic efficiency (Fig. 6a). Control experiments systematically confirmed the essential components of this photocatalytic system: the thiazole-incorporated COF structure, molecular oxygen as oxidant, and appropriate light irradiation all proved indispensable for reaction progression (Supplementary Table 8). These findings highlight how precise structural control over heteroaromatic connectivity in COFs can dramatically influence their photocatalytic performance in aerobic oxidation reactions.

To assess the generality of this photocatalytic system, we examined the oxidation of various ethylbenzene derivatives under standardized conditions (Fig. 6a). TTT-COF maintained exceptional catalytic efficiency for substrates bearing electron-donating halogen substituents (p-F, p-Cl, p-Br), while the electron-withdrawing p-OCH₃ variant showed slightly reduced activity (86% yield). Notably, the framework demonstrated remarkable versatility in converting p-diethylbenzene to the industrially valuable p-diacetylbenzene with 98% yield—a challenging transformation for conventional catalysts. The structural robustness of TTT-COF was evidenced through five consecutive catalytic cycles without significant activity loss (Fig. 6b), with post-reaction PXRD and FTIR analyses confirming complete retention of crystallinity and chemical functionality (Supplementary Figs. 53, 54). This combination of broad substrate scope and exceptional stability positions thiazole-based COFs as practical candidates for sustainable oxidation catalysis.

In order to explore the catalytic mechanism, we first identified the active species in the reaction process. A series of quenching experiments were performed by adding quenching agent to the original test solution of ethylbenzene oxidation (Fig. 6c). When a hole quencher (KI) or electron quencher (AgNO₃) was added, only a trace of the products could be detected after 8 h light irradiation, suggesting that photogenerated electrons and hole radicals are engaged in the reaction. When a ¹O₂ radical quencher (L-Histidine, L-HIS) and •O₂⁻ radical quenching agent (p-benzoquinone, BQ) were introduced, the catalytic efficiency was significantly weakened, stating O₂ is reduced to •O₂⁻ and ¹O, and involved in the oxidation reaction. To verify the generation of the above-mentioned ROS, EPR spectroscopy in MeCN of TTT-COF was performed. As shown in Fig. 6d and Supplementary Fig. 55, •O₂⁻ and ¹O₂ signals appeared in the system after 5 min of illumination, indicating that TTT-COF could rapidly activate oxygen molecules to ROS under light irradiation. However, no ¹O₂ was detected in TTI-COF, which may be the main reason why TTI-COF cannot catalyze ethylbenzene oxidation. According to the obtained results, the reaction mechanism for the photocatalytic aerobic oxidation of C(sp³)-H bonds is proposed in Fig. 6e. The presence of EnT in TTT-COF is attributed to the formation

of homologous heteropolyaromatic structure with thiazole rings, which augment the overall conjugation of the framework and facilitates the formation of $^1\text{O}_2$, thereby facilitating efficient photocatalytic performance toward aerobic oxidation of $\text{C}(\text{sp}^3)\text{-H}$ bonds.”

Fig. 6 The photocatalytic performance of TTT-COF for aerobic oxidation of $\text{C}(\text{sp}^3)\text{-H}$ bonds. **a** TTT-COF catalyzed the aerobic oxidation of ethylbenzene derivatives. **b** Kinetic profile for oxidation of ethylbenzene. Reaction conditions: photocatalyst (10 mg), ethylbenzene (0.1mmol), xenon lamp (420-1100 nm, 15 W), CH_3CN (3 mL), O_2 (1 atm), 8 h. Determined by GC-FID using chlorobenzene as the internal standard, conversion of ethylbenzene, selectivity of acetophenone. **c** Cycle experiment of aerobic oxidation of ethylbenzene along with TTT-COF as photocatalyst. **d** Control experiments for

photocatalytic ethylbenzene oxidation by TTT-COF under normal conditions or with different scavengers. **e** EPR signals of the reaction solution under dark and visible light illumination in the presence of TEMP as the spin-trapping reagent. **f** Proposed mechanism for aerobic oxidation of C(sp³)-H bonds. Experiments were repeated independently (**a-e**) three times with similar results. In **e**, a.u. indicates the arbitrary units. Source data are provided as a Source Data file.

Supplementary Figure 53. PXRD patterns of TTT-COF before and after five photocatalytic oxidation cycles of ethylbenzene (COFs were regenerated by washing with acetone and MeOH), a.u. indicates the arbitrary units.

Supplementary Figure 54. FTIR spectra of TTT-COF before and after five photocatalytic oxidation cycles of ethylbenzene (COFs were regenerated by washing with acetone and MeOH), a.u. indicates the arbitrary units.

Supplementary Figure 55. a EPR spectra of DMPO·OH generated by TTT-COF and TTI-COF in the dark and under visible light irradiation. (300 W Xenon lamp, 5 min irradiation; O₂ saturated; acetonitrile/ethylbenzene mixture (9/1 v/v, 500 μL)), a.u. indicates the arbitrary units.

(13) In Figure 6, TTT-COF was integrated into the polypropylene fiber (PP) and mask, which contains three layers. The PP with COF layer was positioned at the second layer that protected by the other two PP layers and cannot access the light. That is to say, the design is invalid as photocatalysis cannot proceed.

Response to comment:

Thank you for your constructive remarks on our manuscript. We agree that the bacterial disinfection section regarding the creation of a mask is not perfect enough in this manuscript, and, to be honest, we can't explain this mechanism well for now. Considering the suggestions of the editor and the other reviewers, we have decided to delete the contents regarding the creation of a mask to avoid ambiguity. Meanwhile, we further evaluated the photocatalytic performance toward aerobic oxidation of C(sp³)-H bonds of TTT-COFs in order to make the paper focus more on photocatalytic applications.

We thank all referees again for their helpful comments and suggestions, and hope that this significantly revised manuscript is now acceptable for publication in *Nature Communications*.

Best Regards,

Yours Sincerely,

Prof. Dr. Chong Cheng (on behalf of the authors)